# Structural Characterization, Rheology, Texture, and Potential Hypoglycemic Effect of Polysaccharides from *Brasenia schreberi*

**DOI:** 10.3390/foods14101836

**Published:** 2025-05-21

**Authors:** Zhangli Jia, Yin Chen, Chunyu Niu, Yan Xu, Yan Chen

**Affiliations:** College of Food and Pharmacy, Zhejiang Ocean University, 1 South Haida Road, Zhoushan 316000, China; jiazhangli@zjou.edu.cn (Z.J.); mojojo1984@163.com (Y.C.); niuchunyu0808@126.com (C.N.)

**Keywords:** *Brasenia schreberi* polysaccharide, structural characterization, rheology, texture, hypoglycemic effect

## Abstract

*Brasenia schreberi* (BS) is a perennial aquatic plant of the water lily family, of which the recognition as a functional food is on the rise. Polysaccharides from BS have been found to possess antihyperglycemic and antihyperlipidemic activities. This study aimed to partially clarify the structural and evaluate the hypoglycemic potentials of *Brasenia schreberi* polysaccharide (BSP). In this study, BSP was isolated from the mucilage covering the surface of *Brasenia schreberi* (BS). SEM and AFM results verified that BSP molecules were tightly connected and formed a ring-shaped network structure. Further structural analysis showed that BSP was an acidic heteropolysaccharide with a molecular weight of 2.47 × 10^4^ Da. It had 1,2,3-linked α-D-Gal*p*, 1,2-linked α-D-Man*p*, and 1,4-linked β-GlcA residues as the main chain, with 1,3-linked α-Gal*p*, 1,3-linked α-Fuc*p*, 1,3-linked α-Xyl*p*, T-Ara*f*, and T-Rha*p* as side chains. The rheological results indicated that the BSP solution was a pseudoplastic fluid and exhibited shear-thinning properties. Moreover, the gel strength and texture properties of BSP tended to be higher as the BSP and Ca^2+^ concentration increased. More importantly, BSP exhibited good inhibitory activity against α-amylase and α-glucosidase, indicating that it may be a good candidate for a hypoglycemic functional food.

## 1. Introduction

*Brasenia schreberi* (BS) is a perennial aquatic plant of the water lily family, mainly found in Asia, North America, Central America, and Australia. In Asia, it is widely cultivated and traded due to its outstanding value for food and medicinal purposes [1]. The surface of BS sprouts and petioles is covered with a layer of transparent gelatinous mucilage. Studies have shown that the main nutrient components of BS mucilage are acidic polysaccharides, polyphenols, and a small number of proteins, and its polysaccharide component is mainly composed of polymerized monosaccharides such as galactose, fucose, mannose, and glucuronic acid [2]. Since its initial discovery, BS has primarily been investigated in the fields of anatomy, morphology, and physiology. Elakovich et al. [3] found that BS mucilage extract has good biological functions such as antibacterial, algal, and allelopathic effects. This discovery has aroused great attention to the physicochemical properties and biological activity of BS mucilage. It has been reported that BS mucilage is composed of polysaccharide gels with a laminar structure and that the laminar structure of the mucilage and the action of water molecules give it good lubricating properties, so it is often used in the machinery industry [4]. Moreover, *Brasenia schreberi* polysaccharide (BSP) was added to yogurt fermentation by Wang et al. [5]. The results showed that BSP significantly improved the water retention, viscosity, and elasticity of yogurt as well as the viability of lactic acid bacteria. Based on the unique physicochemical and structural characteristics of BS polysaccharide gel, it can be hypothesized that it has great potential for the development of functional foods, so further studies on its microstructure, rheological properties, and textural properties are necessary.

In recent years, there has also been an increasing interest in pharmacological studies of BSP. Previous studies have indicated that BSP has a variety of biological functions such as antioxidant, anti-inflammatory, and cholesterol lowering. It has been explored that BSP has a good scavenging effect on 2,2-Diphenyl-1-picrylhydrazyl (DPPH) and 2,2’-Azino-bis (3-ethylbenzothiazoline-6-sulfonic acid) (ABTS) free radicals, and this antioxidant capacity is closely related to the content of sulfuric radicals and uronic acid in the polysaccharides [6]. In addition, other reports confirmed that BSP could reduce the cholesterol level in hamsters by modulating the expression of genes related to the metabolism of hepatic cholesterol and bile acids [1]. Wan et al. [7] also indicated that polysaccharide hydrogels on the surface of BS leaves were able to prevent ulcerative colitis in mice by regulating intestinal flora. Much research has focused on the antioxidant, anti-inflammatory, hypocholesterolemic, and other bioactivities of BS gel polysaccharides; however, their potential hypoglycemic activities have not been systematically studied. Moreover, numerous studies have demonstrated that plant polysaccharides have favorable hypoglycemic effects, so it is necessary to further investigate the hypoglycemic effects of BSP [8,9].

In this study, BSP was extracted using alkaline extraction and alcohol precipitation from the surface of fresh BS. Then, the microstructure of BSP was observed using SEM and AFM, and further structural analysis such as FT-IR, partial acid hydrolysis, methylation analysis, and NMR analysis was carried out to determine the specific structure of BSP. More importantly, the rheological and textural properties and the potential hypoglycemic effect of BSP were explored. The information obtained in this work will provide theoretical support for the development of hypoglycemic functional foods from BSP.

## 2. Materials and Methods

### 2.1. Materials

Fresh BS was purchased from Hangzhou Chunye Co., Ltd., (Zhejiang, China). Monosaccharide standards were purchased from Sigma-Aldrich Chemical Co., (St. Louis, MO, USA). Trifluoroacetate (TFA), α-Amylase, α-glucosidase, 4-Nitrophenyl-β-D-glucopyranoside (PNPG), and other chemical reagents were purchased from Aladdin Biochemical Technology (Shanghai, China). All other chemical reagents were of analytical grade unless stated otherwise.

### 2.2. Sample Preparation

Fresh BS was shaken at 40 °C for 4 h using a magnetic stirrer to separate the BS gel from the blades, and the supernatant was obtained by centrifugation using a high-speed centrifuge at 7000 r/min for 10 min. Then, BSP was prepared by alkaline extraction and alcohol precipitation: 0.1 M NaOH solution was added to the supernatant in a 1:40 ratio and heated at 30 °C for 4 h. The mixture was centrifuged at 7000 r/min for 10 min and the supernatant was retained; 2 M HCl was added to adjust the pH to neutral, and the extract was concentrated to 1/10. Subsequently, 4-fold pre-cooled 95% ethanol solution was added and left at 4 °C for 12 h, and the precipitate was retained after centrifugation at 7000 r/min for 10 min.

The Sevage method was used to extract the proteins from the polysaccharide solution after the precipitate had been redissolved with purified water: 1/4 volume of Sevage reagent (chloroform: n-butanol = 5:1) was added to the polysaccharide solution. Mix well and stir with a magnetic stirrer for 30 min. Centrifuge the solution at 4000 r/min for 15 min, collect the supernatant, and then continue to add Sevage reagent to remove the protein impurities repeatedly until the intermediate protein layer disappears after centrifugation. Finally, a small amount of Sevage reagent was removed from the supernatant by rotary evaporator [10]. The polysaccharide solution was then dialyzed in running water for 48 h using regenerated cellulose membrane dialysis bags with a molecular weight cut-off of 3500 Da to remove the salt. Finally, the crude polysaccharide was obtained by vacuum freeze-drying the solution for 48 h [11].

The crude polysaccharide was then processed by anion-exchange chromatography on a Q Sepharose Fast Flow column (300 × 30 mm) (GE Healthcare Co., Uppsala, Sweden) using 0.1 M NaCl as the mobile phase. The polysaccharide-containing fraction was collected for further purification with 0.2 M NH_4_HCO_3_ through a Sephacryl S-200 column (2 × 100 cm) (GE Healthcare Co., Uppsala, Sweden). The purified polysaccharide fraction was collected and lyophilized using a vacuum freeze dryer (Beijing Sihuan Scientific Instrument Factory, Beijing, China) (cold trap temperature: −40 °C; vacuum: 0.05–0.08 mba; time: 48 h) [12].

### 2.3. Microstructure Analysis

The microstructure of BSP was observed by SEM and AFM. SEM: the prepared polysaccharide samples were added dropwise on a single crystal silicon wafer and left at room temperature until dry for scanning observation. AFM: an appropriate amount of 10 μg/mL BSP samples was dropped on the surface of freshly peeled mica flakes, dried at room temperature, and then subjected to atomic force microscopy [13].

### 2.4. Molecular Weight Analysis

A high-performance gel permeation chromatography (HPGPC) system consisting of an Agilent 1100 Series refractive index detector and a TSK gel G3000PWXL column (8.0 mm × 30.0 cm, Tosoh, Tokyo, Japan) was used to determine the molecular weight of BSP. Chromatographic conditions: the chromatographic column was SB-804 (8.0 mm × 300.0 mm); the column temperature was set at 35 °C; the injection volume was 20 μL; and 0.02% NaN_3_ solution was used as the mobile phase at a flow rate of 1 mL/min. The standard dextran solutions of different molecular weights and BSP samples were prepared at a concentration of 5 mg/mL, which were filtered through a 0.45 μm microporous filter membrane and analyzed. The standard curve was plotted according to the obtained results, and the molecular weight of BSP was calculated according to the regression equation of the standard curve [14].

### 2.5. Monosaccharide Composition Analysis

BSP (5 mg) was dissolved in 1 mL 2 M trifluoroacetate (TFA) and completely hydrolyzed at 120 °C for 4 h; then, the TFA was evicted by ethanol through rotary evaporation. Monosaccharide composition was measured by high-performance liquid chromatography (HPLC) and a UV detector (Agilent Technologies with 1200 Series detector, Santa Clara, CA, USA) after pre-column derivatization with 1-phenyl-3-methyl-5-pyrazolone (PMP). The standards are composed of L-arabinose (Ara), L-fucose (Fuc), D-galactose (Gal), D-galacturonic acid (GalUA), D-glucose (Glc), N-acetyl-β-D-glucosamine (GlcNAc), D-glucuronic acid (GlcUA), D-mannose (Man), L-rhamnose (Rha), and D-xylose (Xyl). Monosaccharide composition was identified by comparing the peak time between BSP and standard sugars [15].

### 2.6. Partial Acid Hydrolysis

The BSP sample (10 mg) was hydrolyzed with 0.01 M TFA (2 mL) at 100 °C for 1 h. Then, the TFA was eliminated from the sample by rotary evaporation, and the residue was further purified using methanol. Following this, four volumes of anhydrous ethanol were added to the hydrolysate, which was stored at 4 °C overnight. The resultant precipitate and supernatant were designated as P-1 and S-1, respectively, after high-speed centrifugation at 7000 rpm for 10 min. Then, P-1 was sequentially hydrolyzed with 0.05 M, 0.1 M, and 0.5 M TFA at 100 °C for 1 h as previously described. The supernatants obtained in each step and the final precipitates were named S-2, S-3, S-4, and P-4, respectively. After S-1, S-2, S-3, S-4, and P-4 were fully hydrolyzed with 2 M TFA for 3 h at 120 °C, the monosaccharide composition was analyzed using high-performance liquid chromatography (HPLC) with pre-column derivatization using PMP [12].

### 2.7. FT-IR and Methylation Analysis

An amount of KBr powder dried to a constant weight was mixed with polysaccharide samples (BSP:KBr = 1:100) and ground to a powder. It was then pressed into 1 mm transparent tablets using a tablet press. BSP was measured for FT-IR (Tensor, Bruker Optics, Ettlingen, Germany) on Nicolet Omnic 6.0 software and the Nicolet Nexus 470 instrument in the frequency range of 4000–500 cm^−1^ at the resolution of 4.0 cm^−1^ with a background scanning frequency of 32.

### 2.8. Methylation Analysis

Methylation of polysaccharide: 2 mg of vacuum-dried polysaccharide sample was weighed, and 1 mL of DMSO was added to fully dissolve the polysaccharide. The mixture was then transferred to a two-necked flask and placed on a magnetic stirrer for 10 min to ensure that the reactants were completely dissolved to form a homogeneous solution. After that, 100 mg of NaH powder was quickly added and the reaction was stirred at room temperature under nitrogen protection for 1 h. Then, 0.5 mL of CH_3_I was slowly added and the reaction was continued under nitrogen protection and light protection for 1 h. Finally, the reaction was terminated by the addition of 1 mL of ultrapure water. Then, the product obtained from the reaction was extracted by CH_2_Cl_2_ 3 times, and the CH_2_Cl_2_ layer was back-extracted with distilled water 3 times to remove the water-soluble impurities. The collected CH_2_Cl_2_ organic phase was dried at 45 °C.

Hydrolysis of methylated polysaccharides: The methylated product was fully dissolved in 1 mL of 2M TFA solution and transferred to an ampoule. The reaction was sealed and hydrolyzed at 110 °C for 6 h. At the end of the reaction, the TFA was removed by repeated rotary evaporation with methanol and the samples were dried at 45 °C.

Reduction of hydrolyzed products: 1.0 mL of 0.05 M NaOH and 10 mg of NaBH_4_ were added to the acid-hydrolyzed product and mixed thoroughly, then reduced at room temperature for 3–4 h. After the reaction, slowly add glacial acetic acid to the mixture until no gas bubbles are generated in the reaction system. Finally, the boric acid was removed by repeated rotary evaporation with methanol and the samples were dried at 45 °C.

Acetylation of reduced products: add 0.5 mL of pyridine to the reduced products and react in a water bath at 90 °C for 0.5 h. At the end of the reaction, remove the samples and cool them to room temperature, add 0.5 mL of acetic anhydride, and continue the reaction in a water bath at 100 °C for 1 h. The pyridine was removed by repeated rotary evaporation with methanol. Finally, 1.0 mL of CH_2_Cl_2_ was used to dissolve the pyridine and back-extracted three times with ultrapure water to remove insoluble salts and residual pyridine. The lower layer of CH_2_Cl_2_ was collected, dried, and prepared for use.

The products obtained after methylation, hydrolysis, reduction, and acetylation were analyzed by GC-MS (Agilent Technologies Co., Ltd., Santa Clara, CA, USA). According to the GC-MS results and the existing polysaccharide mass spectrometry database (CCRC Spectral Database-PMAA-UGA Database), the connection mode of the glycosidic bond of each component of sea buckthorn polysaccharide can be inferred. By integrating the peak areas of the major peaks in the GC-MS spectra, the percentage of glycosidic bonds of each polysaccharide component in each linkage mode could be known [16].

### 2.9. NMR Spectroscopy Analysis

A 100 mg BSP sample was dissolved in 1 mL D_2_O and freeze-dried repeatedly so that H_2_O in the sample solution was replaced by D_2_O. The replaced sample was dissolved in 0.5 mL D_2_O. ^1^H NMR (size of real spectrum 32,788, spectrometer frequency 600.13, spectrum reference frequency 16.4), HSQC NMR (size of fid 4758-360, number of scans 32, number of dummy scans 16), ^1^H-^1^H COSY (size of fid 5046-355, number of scans 16, number of dummy scans 16), and NOESY (size of fid 7352-512, number of scans 8, number of dummy scans 128) were performed on an Agilent DD2-600 MHz NMR spectrometer (JEOL, Tokyo, Japan) and acquisition of the spectra was carried out using Topspin 2.1.6 software. All spectra were acquired at a temperature of 298 K [17].

### 2.10. Rheological Properties and Texture Analysis

It has been shown that increasing the polysaccharide concentration may lead to a denser network structure, consequently enhancing its viscosity. Additionally, a significant interaction occurred between low concentrations of Ca^2+^ ions and polysaccharides, which contributes to the elevated viscosity, diminished gelling properties, and enhanced thermal stability of the polysaccharides. This phenomenon can be attributed to the ability of Ca^2+^ ions to establish ionic bonds with carboxyl (-COOH) and other functional groups within the polysaccharide molecules, forming what is referred to as a “calcium ion bridge”. This interaction intensifies the connections between polysaccharide molecules, resulting in a more compact and stable gel network structure [18,19,20].

In this study, different concentrations of BSP solutions (2%, 4%, 6%, 8%) were prepared to investigate the effect of concentration on the rheological properties and texture. Additionally, varying concentrations of CaCl_2_ solution (0.5%, 1.0%, 1.5%) were incorporated into the BSP solution (2%) to investigate the effect of different calcium ion concentrations on the rheological properties and texture of BSP [21]. The rheological properties of BSP were determined using an HR-20 rheometer (TA Instruments, New Castle, DE, USA) equipped with parallel plates (40 mm diameter, 1 mm gap). Each group of samples was stirred at 40 °C for 1 h to form a stable and homogeneous gel (30 mm diameter, 50 mm thickness). And the texture was determined using a TAXT-2I texture analyzer (SMS Co., Glasgow, UK) equipped with P/50 probe to obtain the parameters of hardness, cohesion, elasticity, and chewiness of the samples after being stored overnight at 4 °C. The experiment parameters of the texture analyzer were set as follows: the pre-test speed, test speed, and the latter test speed were set at 2.0 mm/s; the time between compressions was 5 s; the strain was 50%; the trigger type was automatic; and the trigger force was 5 g [22]. All tests were carried out in triplicate.

#### 2.10.1. Steady Rheological Properties

A steady shear experiment was carried out at 25 °C with a shear rate range of 0.1–100 s^−1^ and a step time of 60 s. The well-known power-law model, τ = Kγ^n^, was used to fit the data to explain the variation in the rheological properties of samples under steady shear, where n is the dimensionless flow behavior index, K is the consistency coefficient (Pa s^n^), τ is the shear stress (Pa), and γ is the shear rate (1/s) [23].

#### 2.10.2. Dynamic Rheological Properties

The dynamic rheological properties of BSP were assessed at a strain of 2% with a frequency range of 0.1 to 10 Hz. The values of G′ and G′′ were acquired [24].

### 2.11. Hypoglycemic Activity

#### 2.11.1. α-Amylase Inhibition Activity

The α-amylase inhibitory activity of BSP was measured according to the previously described method with minor modifications [25]. Briefly, 1.0 mL of BSP at different concentrations (1.0–3.5 mg/mL) was mixed with 0.5 mL α-amylase solution (0.1 U/mL, dissolved in 0.1 M phosphate buffer with a pH of 6.8), and then the reaction was started by adding 2% starch solution (0.5 mL). The reaction was terminated by adding 3, 5-dinitrosalicylic acid (DNS) reagent (0.2 mL) after incubation at 37 °C for 5 min, and continued to react in a boiling water bath for an additional 5 min. Then, the absorbance value (A_1_) was detected at 540 nm. Acarbose was taken as the positive control. The absorbance value (A_2_) was determined by substituting the enzyme solution with the buffer solution, and the absorbance value (A_3_) was measured by substituting the BSP solution with the buffer solution. The absorbance value (A_0_) was measured by using a buffer solution as a blank group. The inhibition of the enzyme activity was calculated as in Equation (1):(1)$$\mathrm{Inhibition}\;\mathrm{ratio}\;(\%)=\;(1-\frac{A_{1}-A_{2}}{{A_{3}-A}_{0}})\times 100\%$$

#### 2.11.2. α-Glucosidase Inhibitory Activity

The α-glucosidase inhibitory activities of BSP at different concentrations (1.0–3.5 mg/mL) were measured according to the previously described method with minor modifications [26]. Briefly, after adding 0.1 U/mL α-glucosidase (1 mL) to 1 mL of BSP solution, the mixture was placed in a water bath at 37 °C for 10 min. Then, 0.01 M PNPG (0.5 mL) was added and held at 37 °C for 20 min. Subsequently, 0.1 M Na_2_CO_3_ (2 mL) was added to stop the reaction, and its absorbance (A_1_) was measured at 400 nm. Buffer solution was used instead of enzyme solution to measure its absorbance value (A_2_), and buffer solution was used instead of BSP solution to measure its absorbance value (A_3_). The absorbance value (A_0_) was measured by using a buffer solution as a blank group. Acarbose served as the positive control. The enzyme activity inhibition was calculated as in Equation (2):(2)$$\mathrm{Inhibition}\;\mathrm{ratio}\;(\%)=\;(1-\frac{A_{1}-A_{2}}{{A_{3}-A}_{0}})\times 100\%$$

### 2.12. Statistical Analysis

The data are presented as the mean of triplicate determinations unless otherwise specified. Statistical significance was assessed by one-way analysis of variance (ANOVA) and multiple comparison test using SPSS 26.0 software. The level of significance was set at *p* < 0.05. Spectra drawings and line plots were performed using Origin 2018 software. NMR spectra were analyzed and plotted using MestReNova 11.0 software.

## 3. Results and Discussion

### 3.1. Microstructure Analysis

Figure 1a,b illustrate the micro-morphological properties of BSP. As seen in Figure 1a, BSP are smooth sheet-like structures with some irregular spherical structures attached to the surface. Moreover, it can be seen that the structures of BSP are cross-stacked with each other to form a certain reticular structure. This indicates that the molecules of BSP are tightly bound and have strong interactions with each other [27].

In recent years, AFM has become a useful tool for analyzing the surface topography and conformation of high molecular polymers and biomacromolecules. Spheres, random linear chains, and random chains with branches and rods are commonly reported polysaccharide conformations [28]. The 2D and 3D AFM images of BSP are shown in Figure 1b. BSP showed a ring-shaped mesh structure formed by interconnecting macromolecular chains. This was in good agreement with the measurements performed by SEM. Combined with the previous literature, it can be hypothesized that SBP may be a cohesive structure consisting of several molecular chains linked by hydrogen bonding or van der Waals forces [29]. This aggregated structure is likely to have an important effect on the rheological, textural properties, and biological activity of polysaccharide.

### 3.2. Molecular Weight and Monosaccharide Composition Analysis

According to the HPGPC data, the polysaccharide sample’s peak was a single, symmetrical peak (Appendix A), which indicated that BSP was of high purity and could be used for later structural analysis [30]. Based on the standard curve of established molecular weight standards (Appendix A), the molecular weight of BSP was determined to be 2.47 × 10^4^ Da. The biological activity of polysaccharides, including water solubility and cell membrane permeability, is markedly influenced by their molecular weight. Nevertheless, there is no biological action in tiny molecular polysaccharides. Strong biological activity is demonstrated by polysaccharides with molecular weights ranging from 10,000 to 50,000 Da, according to a prior study [31]. The molecular weight of BSP in this study was between 10,000 and about 50,000 Da, which is within the range of bioactive compounds. It can be hypothesized that it may have potential biological roles and its biological activity can be evaluated subsequently.

Analysis of the monosaccharide composition of PMP pre-column derivatization showed that the monosaccharide composition of BSP mainly consisted of Gal, Fuc, and Man, with small amounts of Rha, GlcA, Xyl, and Ara (Appendix A). According to the relative peak area ratio of Man, Rha, GlcA, Gal, Xyl, and Ara, the molar ratio of the monosaccharide components was about 1.0:0.57:0.24:2.73:0.63:0.06:1.73. It can be seen that a small amount of GlcA is present in BSP, and it is hypothesized that BSP may be an acidic polysaccharide. In addition, this result is similar to the monosaccharide composition of the polysaccharide gel coating of the leaves of BS reported by Kim et al. [1]. The percentage of monosaccharides were Gal (25.2%), Man (1.6%), Fuc (6.9%), Rha (7.6%), Ara (3.1%), Xyl (5.5%), Glc (2.1%), and alduronic acids (19.6%). The main monosaccharide component of both was Gal, the difference being that the content of GlcA in the present study was low, whereas the content of alduronic acids in the polysaccharides extracted from the gel coating of the leaves of BS by Kim et al. was second only to that of galactose. This may be caused by the differences in the origin, maturity, and extraction and purification methods of BS raw materials.

### 3.3. FT-IR Analysis

FT-IR analysis is one of the standard techniques used to characterize polysaccharides. The absorption peaks in the infrared spectrum can be used to determine the kind and vibration of functional groups in polysaccharide molecules [32]. The FT-IR of BSP is displayed in Figure 2. The presence of a big signal peak at about 3461 cm^−1^ was caused by the intermolecular O-H stretching vibration, while the peak at about 2932 cm^−1^ represents the presence of the antisymmetric stretching vibration of C-H bending. The absorption peak at 1647 cm^−1^ was associated with the -OH bending vibration. The absorption peak at 1367 cm^−1^ indicated the C-H bending vibration. The peaks at 1076 cm^−1^ and 1030 cm^−1^ were caused by the C-O-C stretching vibration, which was characteristic of carbohydrates. In addition, the specific peak at 773 cm^−1^ represented the planar rocking vibration of CH_2_ [33]. The above results are typical of the functional group information possessed by polysaccharides in much of the relevant literature on FT-IR analysis [12,17]. More importantly, the band at 1730 cm^−1^ confirmed the existence of uronic acids [34]. From these results, it can be hypothesized that BSP is an acidic polysaccharide. This was relatively consistent with the results for monosaccharide composition.

### 3.4. Partial Acid Hydrolysis

HPLC was used to determine the monosaccharide composition of BSP. However, as a polysaccharide with complex monosaccharide composition, the distribution of monosaccharides needs to be further processed by partial acid hydrolysis. This approach is used to analyze the structure of polysaccharides by taking advantage of the principle that branched glycosidic bonds in polysaccharide chains are easily hydrolyzed while the main chain part is relatively stable [35].

Table 1 displays the monosaccharide compositions of the five fractions S-1, S-2, S-3, S-4, and P-4 which were mostly produced by partial acid hydrolysis of BSP. The results showed that S-1 and S-2 were mainly composed of Fuc, Gal, Ara, Rha, and a small amount of Man. Their contents increased remarkably as the acid concentration increased, indicating that these monosaccharides are located in the outermost branched chain of BSP and sensitive to acid conditions. Moreover, Xyl was degraded down, indicating that Xyl was located in the inner position of the branched chain. The amount of Rha, Ara, and Fuc in S-3 started to decline following the third hydrolysis with an even higher acid concentration, while the contents of Gal and Xyl still increased, indicating that these two monosaccharides also existed close to the position of the main chain. The main components of Gal and Man as well as a tiny quantity of GlcA in S-4 and P-4 revealed that the main chain of BSP mostly consisted of these several kinds of monosaccharides. Gal and Man are always present in these five components, indicating that Man and Gal are present in both the branched chain and the main chain. GlcA was only present in S-4 and P-1, indicating that GlcA is most likely present in the main chain [36]. The above results indicated that BSP has a complex branched chain structure.

### 3.5. Methylation Analysis

To determine the linkages of the glycosidic bonds and to further understand the structure of BSP, methylation and GC-MS investigations were carried out. The results showed that BSP had a complex structure mainly containing seven different linkage styles (Table 2). Gal residues mainly had 1,3-Gal*p* and 1,2,3-Gal*p* linkages with a 2:1 ratio; Man was linked as 1,2-Man*p*; Xyl had a 1,3-Xyl*p* linkage; Fuc had a 1,3-Fuc*p* linkage; and the Ara and Rha residues existed as terminal Ara*f* and terminal Rha*p* linkages, respectively. According to previous reports, BSPs were mainly polysaccharides with Gal and Man as backbones. From the methylation analysis combined with the data of stepwise acid hydrolysis, BSP may have a similar feature, with 1,3-Ga*lp*, 1,2,3-Gal*p*, and 1,2-Man*p* as the main chains, and 1,3-Fuc*p,* 1,3-Xyl*p*, terminal Ara*f*, and terminal Rha*p* as the side chains [4]. However, the specific linkage of GlcA and the sequence of these glycosidic bonds need to be further investigated.

### 3.6. NMR Spectroscopy Analysis

NMR spectroscopy is currently one of the most effective and accurate methods for polysaccharide structure identification [37]. To further elucidate the structure of BSP, it was analyzed by one-dimensional (1D) ^1^H-NMR and ^13^C-NMR spectra and two-dimensional (2D) ^1^H-^1^H correlation spectroscopy (^1^H-^1^H COSY), heteronuclear single quantum correlation spectroscopy (HSQC), and nuclear Overhauser effect spectroscopy (NOESY). One-dimensional NMR (^1^H and ^13^C NMR) was commonly used to determine the configuration of the glycosidic bond (α- or β-configuration) as well as the type and proportion of polysaccharide residues. HSQC shows the correlation between protons and directly attached carbon [38]. From the ^1^H and ^13^C spectra as well as HSQC spectra (Figure 3a–c), it was clear that BSP mainly contains eight residues (residues A, B_1_, B_2_, B_3_, C_1_, C_2_, D_1_ and D_2_) with the anomeric signals H1/C1 at δ 5.23 (98.75), 5.15 (100.60), 5.10 (101.43), 5.07 (100.93), 4.94 (102.31), 4.93 (108.19), 4.32 (103.39), and 4.24 (103.98), where the chemical shifts at D_1_ (δ 4.32) and D_2_ (δ 4.24) were attributed to the β-configuration and the remaining six residues belonged to the α-configuration of sugar residues [38].

^1^H-^1^H COSY reflects the correlation of adjacent protons in the sugar residues, starting from an anomeric proton and progressively correlating the spins around the spin-coupled network, and the assignment of protons to residues can be accomplished by scalar concatenation [38]. The remaining proton signals on the same spin system can be attributed through the COSY spectrum (Figure 3d). The carbon signals associated with polysaccharide residues can be further attributed by HSQC spectroscopy (Figure 3c). Finally, the H/C chemical signals of all sugar residues were collated and summarized as shown in Table 3. These residues can be assigned based on methylation analysis results and information from related publications. A and B_1_ residues were deduced to be 1,3-linked α-D-Gal*p* and 1,2,3-linked α-D-Gal*p*, respectively; residue B_2_ could be a Fuc residue because of its *O*-6 methyl group at 1.18/16.77. Its C3 shifted to 77.13 ppm, indicating the →3)-α-D-Fuc*p*(1→ linkage [39]; B_3_ was determined to be 1,2-linked α-D-Man*p* [17,40]; and C_1_ also had a methyl group and anomeric proton at 4.94 ppm, which could be assigned to T-α-L-Rha*p* [41]. The anomeric carbon of C_2_ was at 108.19, which was located in a low field region and could be characteristic of α-L-Ara*f*. Based on the reference, it was deduced to be T-α-L-Ara*f* [38]; D_1_ with H2 at 3.25 ppm and C3 at about 77 ppm was assigned to be 1,3-linked β-D-Xyl*p* [38]; D_2_ with C6 at 175 ppm was 1,4-linked β-GlcA [41].

The NOESY spectra were able to determine the sequence of the polysaccharide residues [12]. From the NOESY spectrum (Figure 3e), H1 of residue A and H2 of residue B_1_ showed a correlation, indicating that (1→3)-α-D-Gal*p* was linked to the *O*-2 position of (1→2,3)-α-D-Gal*p*. The correlation between H1 of residue B_1_ and H4 of residue D_2_ indicated that the *O*-4 positions of (1→2,3)-α-D-Gal*p* and (1→4)-β-D-GlcA were linked together. H1 of residue B_2_ was associated with H3 of A, H2 of B_1_, and H2 of B_3_, indicating that (1→3)-α-D-Fuc*p* was linked to the *O*-3 position of (1→3)-α-D-Gal*p*, the *O*-2 position of (1→2,3)-α-D-Gal*p*, and (1→2)-α-D-Man*p*, respectively. The correlation between H1 of residue B_3_ and H3 of residue D_1_ indicated that residue (1→2)-α-D-Manp was attached at the *O*-3 of residue (1→3)-β-D-Xyl*p*. H1 of residue C1 was associated with H2 of residue B_1_, and H1 of residue C_2_ was associated with H3 of residue B_2_, indicating that the terminal T-α-L-Rhap (1→ was attached to the *O*-2 position of (1→2,3)-α-D-Gal*p* and the terminal α-L-Ara*f* (1→ was attached to the *O*-3 position of (1→3)-α-D-Fuc*p*. H1 of residue D_1_ was correlated to H3 of A and H3 of B_2_, respectively, indicating that residue (1→3)-β-D-Xyl*p* was connected to the *O*-3 position of (1→3)-α- D -Gal*p* and the *O*-2 position of (1→2)-α-D-Man*p*, respectively.

These NMR analyses combined with the results of stepwise acid hydrolysis and methylation suggested that BSP may have a main chain consisting of (1→2)-α-D-Man*p*, (1→2,3)-α-D-Gal*p*, and (1→4)-β-D-GlcA, with the branching point located at the *O*-2 position of (1→2,3)-α-D-Gal*p*, and with the side chains substituted by (1→3)-α-D-Gal*p* substitution. The outermost layer may consist of (1→3)-α-D-Fuc*p*, terminal α-L-Rha*p* (1→, (1→3)-β-D-Xyl*p*, and terminal α-L-Ara*f* (1→. The possible structure of BSP was deduced according to the information above, as shown in Figure 3f.

### 3.7. Rheological Properties

#### 3.7.1. Steady Rheological Properties

The steady shear rheological curves of BSP are shown in Figure 4a,b. It is evident from the figure that the apparent viscosities of BSP with different concentrations as well as with the addition of different contents of CaCl_2_ solutions declined with the increase in shear rate (Figure 4a). Moreover, the change curve of shear stress with the shear rate was a convex function (Figure 4b), which indicated that BSP solutions are pseudoplastic fluids in non-Newtonian fluids and exhibit shear-thinning characteristics [21]. The exceptional pseudoplastic properties of BSP suggest its potential application in food processing, where it can enhance the viscosity and stability of various food products.

As shown in Figure 4a, the apparent viscosity of BSP rises with increasing concentration at the same shear rate. This could be because the distance between polysaccharide molecular chains decreases and the interaction is enhanced with the increase in polysaccharide concentration, forming a more tightly wound network structure. Moreover, there are a large number of hydroxyl groups in BSP molecules, which have the potential to form intramolecular hydrogen bonds with each other or intermolecular hydrogen bonds with water molecules, and an increase in concentration leads to enhanced interactions and a more stable network structure, thus increasing the apparent viscosity of polysaccharide. This finding is consistent with the study reported by Wang et al. [42]. Furthermore, the apparent viscosity of BSP increased with increasing Ca^2+^ at the same shear rate. This may be due to the bridging effect between Ca^2+^ and polysaccharide molecules, which makes the polysaccharide reticular cross-linking structure more stable, thus increasing the apparent viscosity of polysaccharide. This result is similar to the related report on *Plantago asiatica* L. polysaccharides by Yin et al. [18,19,20].

The parameters of the power-law model that was utilized to describe the flow characteristic curves of BSP are shown in Table 4. All of the coefficients of determination (R^2^) were found to be between 0.98 and 0.99, suggesting that the power-law model adequately described the rheological characteristics of polysaccharides. Furthermore, the fluid’s proximity to the Newtonian fluid is indicated by the flow property index (n), where n = 1 denotes the Newtonian fluid. The lower the value of n, the more pseudoplastic the fluid [23]. As shown in Table 4, n values of all polysaccharide curves are less than 1.0, indicating that they show pseudoplasticity. Moreover, the higher concentration of polysaccharides and Ca^2+^ exhibited a lower n value, which indicated that the increase in polysaccharide concentration and the addition of Ca^2+^ resulted in enhanced pseudoplastic properties for BSP.

#### 3.7.2. Dynamic Rheological Properties

When examining the viscoelasticity of polymer polymeric materials, the energy storage modulus (G’) and loss modulus (G”) are frequently employed metrics. The samples mainly exhibit liquid-like viscosity when G’ < G”, while they mainly exhibit solid-like elasticity when G’ > G” [43]. The variation of G′ and G′′ with frequency for different concentrations of BSP solution (2%, 4%, 6%, 8%) and for BSP solution (2%) with different concentrations of Ca^2+^ solution (0.5%, 1.0%, 1.5%) added is shown in Figure 4c. At frequencies between 0.1 and 10 rad/s, both G’ and G” of BSP rose as the concentration of Ca^2+^ addition and frequency increased.

As can be seen in Figure 4c, the polysaccharide at a concentration of 2% *w*/*v* was mainly viscous (G’ < G”). There was an intersection A (3.27,15.75) between G’ and G” when the polysaccharide concentration was 4% *w*/*v*. The polysaccharide exhibited viscoelastic properties, demonstrating viscosity at low frequencies (G’ < G”) and elasticity at high frequencies (G’ > G”). The polysaccharides were predominantly elastic (G’ > G”) as the concentration increases to 6% *w*/*v* and 8% *w*/*v*. Moreover, the polysaccharides showed viscosity (G’ < G”) at low frequencies and elasticity (G’ > G”) at high frequencies at Ca^2+^ additions of 0.5% *w*/*v* and 1.0% *w*/*v*. And there were intersections B (5.85,19.13) and C (1.20,16.06) between G’ and G”. It can be seen that the polysaccharide solution with Ca^2+^ addition of 1.0% *w*/*v* changed from viscosity to elasticity earlier during the frequency scanning process. BSP was mainly dominated by elasticity (G’ > G”) when the Ca^2+^ content was increased to 1.5% *w*/*v*. The results showed that the increase in the concentration of polysaccharides and Ca^2+^ could facilitate the formation of a more compact network structure between BSP molecules, and promote the transition of polysaccharides from liquid to gel [44]). The outcomes aligned with the steady rheological characteristics. Its exceptional viscoelasticity presented significant opportunities for future applications in food processing.

### 3.8. Textural Properties

Textural properties are important features for exploring the structure and texture of gelatinous foods, which mainly include indicators such as hardness, elasticity, adhesion, cohesion, and chewiness [45]. The textural properties for different concentrations of BSP solution (2%, 4%, 6%, 8%) and for the BSP solution (2%) with different concentrations of Ca^2+^ solution (0.5%, 1.0%, 1.5%) added are shown in Table 5. The parameters of BSP, such as hardness, elasticity, adhesion, cohesion, and chewiness, increased with the increase in BSP concentration as well as Ca^2+^, which further verified that the increase in polysaccharide as well as Ca^2+^ concentration promoted the intermolecular tight junctions and made the reticular structure more solid. The results are consistent with the rheological properties of polysaccharides described above. The unique structure of BSP endows it with excellent texture properties, which lays a good foundation for its future use in food processing.

### 3.9. Hypoglycemic Activity

α-amylase and α-glucosidase are enzymes that catalyze the hydrolysis of glycogen and starch in the saliva, pancreas, and mucous membrane cells of the small intestine. The α-amylase and α-glucosidase inhibitors interact with α-amylase and α-glucosidase in the body, respectively, to prevent the conversion of glycogen and starch into glucose, thereby effectively controlling blood glucose levels [8,9]. Figure 5a,b display the inhibitory effects of BSP on α-amylase and α-glucosidase. The inhibitory effects on both α-amylase and α-glucosidase tended to rise as the concentration of BSP was gradually raised from 0.25 mg/mL to 5 mg/mL. The inhibitory effect could not change much when the concentration reached 5 mg/mL, indicating that the inhibitory activity of BSP on the enzymes showed an obvious dose–effect relationship within a certain concentration range. The half inhibition rate (IC_50_) values of the inhibitors can be used to evaluate the inhibition efficiency of the inhibitors. The IC_50_ values of acarbose and BSP for α-amylase were 1.046 mg/mL and 2.136 mg/mL, respectively, and for α-glucosidase, they were 0.830 mg/mL and 1.551 mg/mL, respectively, which showed that the inhibition effect of BSP on α-glucosidase and α-amylase with significant inhibitory activity was marginally less than that of acarbose. Therefore, it is concluded that BSP has a hypoglycemic effect in vitro and may be an excellent candidate for hyperglycemic inhibitors [46,47].

## 4. Discussion

This study comprehensively reveals the structural, rheological, and textural properties of BSP and its potential as a functional food component for hypoglycemia. Structural characterization showed that BSP is an acidic heteropolysaccharide whose molecules form a unique ring network structure through tight junctions. This structural feature observed by SEM and AFM is consistent with previous descriptions of the lamellar gel nature of BS mucilage, and this structural property confers lubricating properties and potential for industrial applications [4]. The molecular weight of BSP (2.47 × 10^4^ Da) and its monosaccharide composition including Gal, Man, GlcA, Fuc, Xyl, Ara, and Rha are consistent with earlier findings on *Brasenia schreberi* polysaccharides, but the present study further clarifies its unique sugar chain linkage pattern [2]. These structural features, especially the presence of glucuronic acid residues, may be closely related to their biological activities such as antioxidant and hypoglycemic effects [6].

The pseudoplastic fluid behavior and shear-thinning properties of BSP solutions observed in rheological tests are typical of many polysaccharide gels, suggesting their potential application in food systems such as beverages or dairy products where viscosity modulation is required. The enhancement of the gel strength and textural properties of BSP with increasing concentration and Ca^2+^ concentration reveals the role of divalent ions in stabilizing the structure of the polysaccharide network through ionic cross-linking. This finding is in agreement with Yin et al. [18,19,20]. The rheological and textural properties of BSP make it a multifunctional food ingredient that can enhance both the sensory quality and functional performance of food. For example, Wang et al. demonstrated that BSP can improve the textural and water-holding properties of yogurt through the formation of a stable polysaccharide–protein matrix [5].

The most important finding of this study is the inhibitory effect of BSP on α-amylase and α-glucosidase, which are key targets for postprandial glucose absorption. This suggests that BSP may provide a natural alternative to synthetic α-glucosidase inhibitors such as acarbose, which are often accompanied by gastrointestinal side effects, by delaying carbohydrate digestion and reducing blood glucose fluctuations. The hypoglycemic mechanism of BSP may be related to its structural complexity. For example, the molecular weight, monosaccharide composition, glycosidic bond type, and higher structure of polysaccharides affect their hypoglycemic activity. Generally speaking, lower molecular weight polysaccharides are more likely to bind to enzymes and thus exert better hypoglycemic effects, and polysaccharides containing specific monosaccharides such as mannose and arabinose, as well as specific types of glycosidic bonding and linkages, have stronger hypoglycemic activity. In addition, the functional groups of polysaccharides such as glyoxalate and sulfate groups can enhance their hypoglycemic activity [48].

Despite the positive results of the in vitro enzyme inhibition experiments, the hypoglycemic mechanism of BSP needs to be further investigated by in vivo animal experiments. Unlike synthetic drugs, the in vitro activity of plant polysaccharides is usually mild, but their in vivo efficacy may be achieved through multiple pathways, such as modulation of intestinal flora or improvement of insulin sensitivity. Follow-up studies are needed to validate the in vivo efficacy of BSP in animal models of diabetes and to elucidate their effects on glucose transport, insulin signaling pathways, and intestinal flora [49].

## 5. Conclusions

In this study, an acidic and branched heteropolysaccharide BSP was extracted from BS. The microstructure analysis of BSP revealed that BSP molecules were tightly interconnected, forming a ring-shaped network structure. Further structural analysis indicated that the polysaccharide’s main chain comprised 1,3-linked α-D-Galp, 1,2-linked α-D-Manp, and 1,4-linked β-GlcA residues. The side chains are attached to the O-2 position of 1,3-linked α-D-Galp. The inner part of the side chains contained 1,3-linked α-D-Galp, while the outer part of the side chains had 1,3-linked α-Fucp, 1,3-linked α-Xylp, terminal Araf, and terminal Rhap. The rheological results demonstrate that BSP behaves as a pseudoplastic fluid, exhibiting shear-thinning characteristics. Texture analysis revealed that the hardness, elasticity, adhesion, cohesion, and chewiness of the polysaccharide increased with higher concentrations of BSP and Ca^2+^. Moreover, its ability to inhibit α-amylase and α-glucosidase, combined with its natural origin and low toxicity, positions it as a viable candidate for developing functional foods aimed at diabetes management. In the future, we can focus on the effects of BSP on the rheological properties and textural properties of functional foods, to obtain foods with better taste, richer nutrition, and specific biological activities. In addition, the molecular mechanism of hypoglycemia of BSP needs to be further investigated so that it can be widely used in the clinical treatment of diabetes and the development of hypoglycemic functional foods.

## Figures and Tables

**Figure 1 foods-14-01836-f001:**
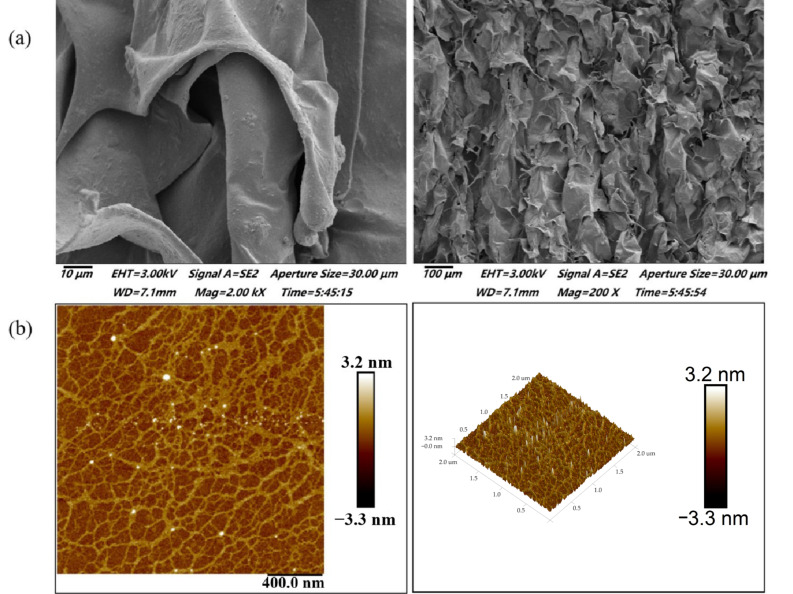
Microstructure of BSP. (**a**) SEM observation of BSP. (**b**) AFM observation of BSP.

**Figure 2 foods-14-01836-f002:**
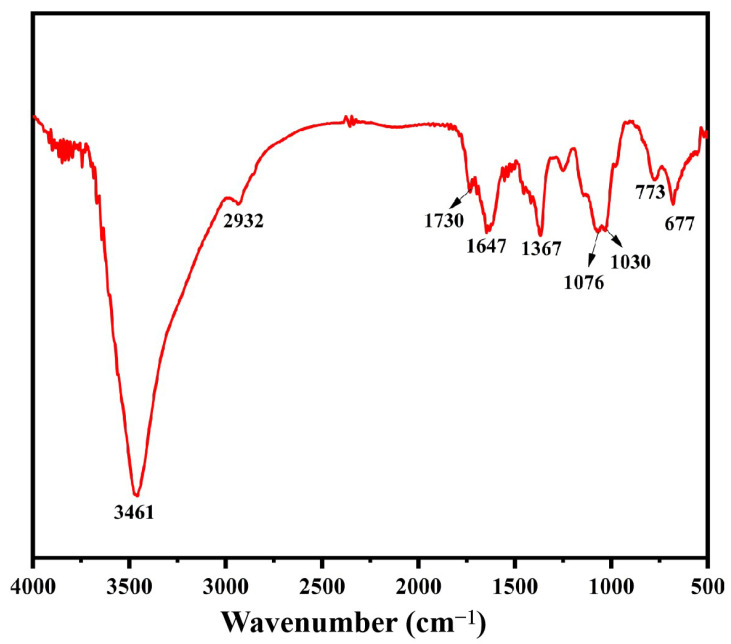
FT-IR spectroscopy of BSP.

**Figure 3 foods-14-01836-f003:**
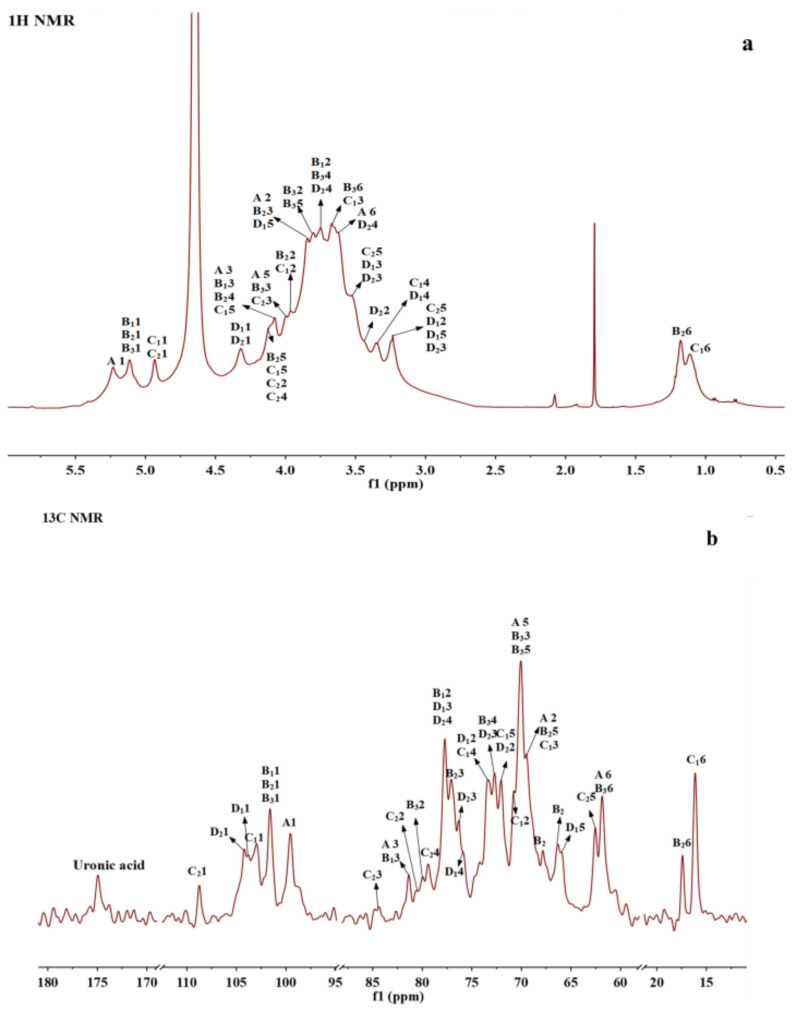
NMR spectra and the structure of BSP. (**a**) ^1^H NMR. (**b**) ^13^C NMR. (**c**) ^1^H-^1^H COSY. (**d**) ^1^H-^13^C HSQC. (**e**) ^1^H-^1^H NOESY. (**f**) The proposed structure of BSP.

**Figure 4 foods-14-01836-f004:**
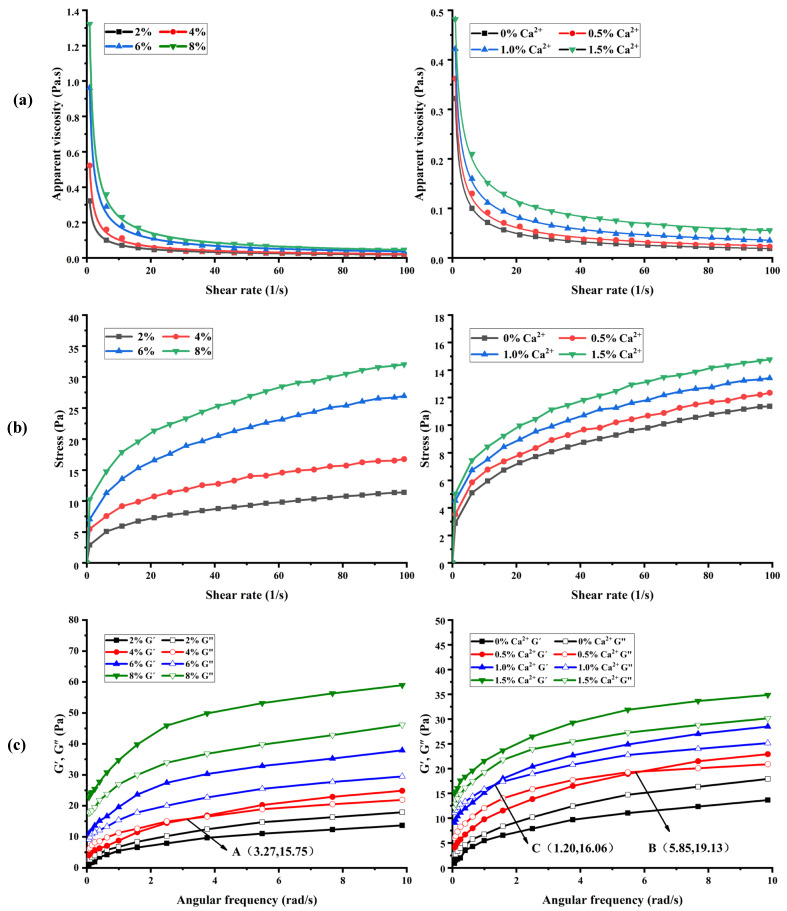
The steady shear rheological and dynamic rheological property curves of BSP. (**a**) The apparent viscosities of BSP with different concentrations and Ca^2+^ additions. (**b**) Effects of shear rate on shear stress of BSP with different concentrations and Ca^2+^ additions. (**c**) Effect of BSP with different concentrations and Ca^2+^ additions on G′, G″.

**Figure 5 foods-14-01836-f005:**
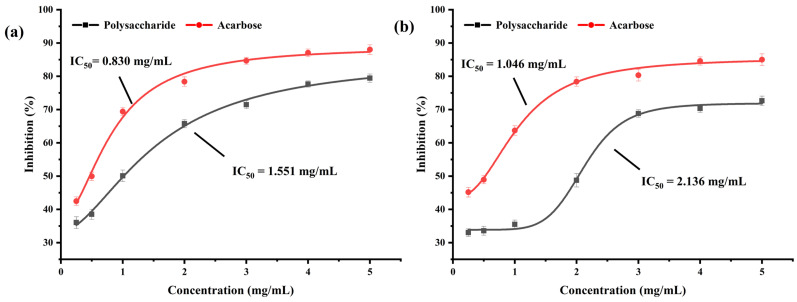
The inhibitory activities of BSP on α-glucosidase (**a**) and α-amylase (**b**).

**Table 1 foods-14-01836-t001:** Monosaccharide composition of the acid-hydrolyzed fraction of BSP.

Sample	Monosaccharide Composition (% m/m)						
	Man	Rha	GlcA	Gal	Xyl	Ara	Fuc
S-1	3.04	10.99	-	19.41	-	18.24	48.31
S-2	2.00	13.24	-	25.95	4.28	12.34	42.19
S-3	2.06	5.58	-	55.89	15.03	-	21.44
S-4	26.87	-	15.28	57.85	-	-	-
P-4	51.37	-	28.48	20.15	-	-	-

**Table 2 foods-14-01836-t002:** GC-MS data analysis of BSP.

Methylated Alditol Acetate	Linkage Type	Main MS (*m/z*)	Molar Ratio (100%)
1,4-Di-O-acetyl-1-deuterio-2,3,5-tri-O-methyl-D-arabinitol	α-L-Ara*f*-(1→	101, 117, 161	1.01
1,5-Di-O-acetyl-1-deuterio-6-deoxy-2,3,4-Tri-O-methyl-L-mannitol	α-L-Rha*p*-(1→	101, 117, 131, 145, 161, 173	8.26
1,3,5-Tri-O-acetyl-1-deuterio-2,4-di-O-methyl-D-xylitol	→3)-Xyl*p*-(1→	101, 117, 129, 161, 173, 189	9.43
1,3,5-Tri-O-acetyl-1-deuterio-6-deoxy-2,4-di-O-methyl-L-galactitol	→3)-Fuc*p*-(1→	101, 117, 131, 155, 161, 207, 233	25.47
1,3,5-Tri-O-acetyl-1-deuterio-2,4,6-tri-O-methyl-D-galactitol	→3)-Gal*p*-(1→	101, 117, 129, 161, 173, 233	17.24
1,2,5-Tri-O-acetyl-1-deuterio-3,4,6-tri-O-methyl-D-mannitol	→2)-Man*p*-(1→	101, 129, 161, 190	15.02
1,2,3,5-Tetra-O-acetyl-1-deuterio-4,6-di-O-methyl-D-galactitol	→2,3)-Gal*p*-(1→	101, 117, 129, 145, 161, 189, 233	23.57

**Table 3 foods-14-01836-t003:** ^1^H and ^13^C NMR chemical shifts (δ) for the residues A_1_-D_2_ of BSP.

Residue	H1/C1	H2/C2	H3/C3	H4/C4	H5/C5	H6/C6
A →3)-α-D-Gal*p* (1→	5.23/98.75	3.82/69.31	4.08/80.75	-	4.03/70.60	3.62/61.07
B_1_ →2,3)-α-D-Gal*p* (1→	5.15/100.75	3.71/77.13	4.04/80.05	-	-	-
B_2_ →3)-α-D-Fuc*p* (1→	5.10/101.03	3.96/69.75	3.85/76.50	4.10/67.12	4.11/69.16	1.18/16.77
B_3_ →2)-α-D-Man*p* (1→	5.07/100.93	3.78/78.90	3.97/70.24	3.72/71.91	3.80/70.01	3.72/61.30
C_1_ T-α-L-Rha*p*	4.94/102.31	3.95/70.78	3.68/68.88	3.35/72.01	4.11/71.16	1.11/15.83
C_2_ T-α-L-Ara*f*	4.93/108.19	4.13/79.18	4.01/83.57	4.10/80.75	3.55/61.84	
D_1_ →3)-β-D-Xyl*p* (1→	4.32/103.39	3.25/75.66	3.62/76.93	3.35/75.56	3.87(3.22)/65.18	
D_2_ →4)-β-D-GlcA (1→	4.24/103.98	3.42/71.41	3.53/76.29	3.69/77.14	-	-/175.00

**Table 4 foods-14-01836-t004:** The flow characteristic of BSP with different concentrations and Ca^2+^ additions.

Concentration		n	R^2^
BSP	2%	0.2975 ± 0.002 ^a^	0.9980
	4%	0.2646 ± 0.003 ^b^	0.8933
	6%	0.2178 ± 0.004 ^c^	0.8865
	8%	0.1748 ± 0.003 ^d^	0.8610
Ca^2+^	0.0%	0.2975 ± 0.002 ^a^	0.9980
	0.5%	0.2741 ± 0.001 ^b^	0.9983
	1.0%	0.2507 ± 0.002 ^c^	0.9970
	1.5%	0.2477 ± 0.002 ^d^	0.9975

All data represent the mean of triplicates. Significant differences (*p* < 0.05) are indicated by different letters within a column.

**Table 5 foods-14-01836-t005:** The textural properties of BSP with different concentrations and Ca^2+^ additions.

Concentration		Hardness (N)	Springiness	Adhesiveness (N·S)	Cohesiveness	Chewiness (N)
BSP	2%	13.51 ± 0.03 ^d^	0.41 ± 0.02 ^d^	18.09 ± 0.05 ^d^	0.75 ± 0.01 ^c^	4.15 ± 0.03 ^d^
	4%	16.23 ± 0.05 ^c^	0.54 ± 0.01 ^c^	25.37 ± 0.03 ^c^	0.78 ± 0.02 ^c^	6.84 ± 0.03 ^c^
	6%	21.18 ± 0.05 ^b^	0.68 ± 0.03 ^b^	31.52 ± 0.04 ^b^	0.83 ± 0.02 ^b^	11.95 ± 0.04 ^b^
	8%	28.26 ± 0.04 ^a^	0.82 ± 0.02 ^a^	36.25 ± 0.05 ^a^	0.87 ± 0.01 ^a^	20.16 ± 0.03 ^a^
Ca^2+^	0.0%	13.51 ± 0.03 ^d^	0.41 ± 0.02 ^d^	18.09 ± 0.05 ^d^	0.75 ± 0.01 ^c^	4.15 ± 0.03 ^d^
	0.5%	15.72 ± 0.04 ^c^	0.46 ± 0.02 ^c^	19.86 ± 0.04 ^c^	0.76 ± 0.02 ^c^	5.50 ± 0.02 ^c^
	1.0%	18.43 ± 0.05 ^b^	0.50 ± 0.03 ^b^	21.74 ± 0.03 ^b^	0.78 ± 0.01 ^b^	7.19 ± 0.03 ^b^
	1.5%	20.29 ± 0.03 ^a^	0.53 ± 0.02 ^a^	23.13 ± 0.05 ^a^	0.80 ± 0.02 ^a^	8.60 ± 0.03 ^a^

All data represent the mean of triplicates. Significant differences (*p* < 0.05) are indicated by different letters within a column.

## Data Availability

The original contributions presented in this study are included in the article/Appendix A. Further inquiries can be directed to the corresponding authors.

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
