# Peer review of "Structural Characterization, Rheology, Texture, and Potential Hypoglycemic Effect of Polysaccharides from Brasenia schreberi"

_foods, 2025, doi:10.3390/foods14101836_

Round 1

Reviewer 1 Report

Comments and Suggestions for Authors

This study examines the extraction and structural properties of an acidic, branched heteropolysaccharide from Brasenia schreberi. This polysaccharide forms a ring-shaped network and consists of a complex arrangement of sugar residues. It exhibits shear-thinning behavior and its textural properties, such as elasticity and adhesion, increase with higher concentrations. The polysaccharide also shows potential for diabetes management due to its ability to inhibit key enzymes.

Despite some interesting results, the manuscript presents some relevant flaws. First of all, some recent studies already assessed the potential of BS-derived polysaccharides as anti-diabetic agents [https://doi.org/10.1016/j.crfs.2022.09.001; https://doi.org/10.1016/j.ijbiomac.2019.05.129]. So I have some concerns about the novelty of the present work under this perspective. Why is this study original? How does it contribute to the literature in the field?

Even the structural investigation of the extracted polysaccharide was already reported as stated by the authors so, again, the novelty of the study is rather obscure to me.

The English must be improved, there are some paragraphs that are very difficult to read. See for example line 45, line 120 (“rotary evaporated” is really unusual), lines 250-252 and so on.

Below I report a list of other several points that must be addressed:

Line 36: The term “natural active substance”, which is repeated several times in the manuscript, is not correct and ambiguous: probably the authors mean “biologically active natural products” or “biologically active substances originating from natural sources”.

Line 37: Bibliographic references are needed

Line 38: BS abbreviation is explained only in the abstract. Please define the abbreviation here as well.

Line 52: Define the BSP abbreviation

Line 102: Really “NaCl2” as the mobile phase? I hope it is a typo. We are all academics and it is quite unacceptable….  Moreover, the concentration is missing.

Line 118-131:  This paragraph should be rewritten and synthetized. It is full of repetitions and it is difficult to read.

Line 133: “An amount” ? It is quite vague.

Line 135: The [21] reference is not necessary, the authors must indicate the model of the IR instrumentation.

Line 142: A very small amount of H2O remains even after several cycles. So I would not say “completely removed”

Line 144: Some information about the pulse sequences must be added.

Lines 148-158: Lots of repetitions in this paragraph, please rephrase it to improve readability. Moreover, why do you select Ca2+ ? You must provide an adequate explanation.

Lines 170-178: This paragraph looks like a laboratory report. It must be modified.

Lines 201-202: Not clear

Line 203: “the molecular chain size of polysaccharides is 0.1-1.0 nm” how do you obtain these dimensions?

Line 221: The biological action is potential and must be assessed, here you take it for granted.

Line 228: Reported where? And if it is already reported this information doesn’t add anything new.

Line 387: In Table 4, as well in Figure 5 and in the discussion of the rheological results is not clear the polysaccharide concentration in the samples containing Ca2+.

Lines 348 – 412: The discussion about the rheological properties of the system should be consistently improved. First of all, all the considerations made are pointless if the concentration of BSP in the samples containing Ca2+ is not reported. If you look at the graphs in Figure 5, in the presence of Ca2+ the viscosities, G’ and G’’ values are generally lower if compared to the pristine BSP samples. Thus, the mentioned structuring effect of calcium ions cannot be really evaluated if you don’t compare all the rheological results before and after the addition of Ca2+ at the same BSP concentration.

Moreover “Viscosity can also be measured using the value of K determined by the power law model; the greater the value of K, the higher the viscosity” is not completely true: this statement holds only if the value of n remains constant across the samples. And in your case, n changes from sample to sample. So, this must be changed.

The viscoelastic response of your system is interesting, but as I mentioned before the discussion must be improved by making a direct comparison between the samples at the same BSP concentration before and after calcium addition. I suggest to discuss also the modifications in the crossover point between G’ and G’’ by adding a graph or a table column with the crossover values.

Also for the textural properties the discussion must be improved as indicated before.

In conclusion, my recommendation is that major revisions are required.

Author Response

Reviewer #1: This study examines the extraction and structural properties of an acidic, branched heteropolysaccharide from Brasenia schreberi. This polysaccharide forms a ring-shaped network and consists of a complex arrangement of sugar residues. It exhibits shear-thinning behavior and its textural properties, such as elasticity and adhesion, increase with higher concentrations. The polysaccharide also shows potential for diabetes management due to its ability to inhibit key enzymes.

Despite some interesting results, the manuscript presents some relevant flaws. First of all, some recent studies already assessed the potential of BS-derived polysaccharides as anti-diabetic agents [https://doi.org/10.1016/j.crfs.2022.09.001; https://doi.org/10.1016/j.ijbiomac.2019.05.129]. So I have some concerns about the novelty of the present work under this perspective. Why is this study original? How does it contribute to the literature in the field?

Even the structural investigation of the extracted polysaccharide was already reported as stated by the authors so, again, the novelty of the study is rather obscure to me.

We sincerely thank the reviewer for the insightful and valuable suggestions, as well as your attention to prior work in this field. In response to your feedback, we’d like to highlight how our study adds fresh perspectives and drives progress in the domain.

Compared to existing studies, our work stands out in several ways: (1) Different Brasenia schreberi raw materials and extraction methods may affect the structural characteristics of polysaccharides and their biological activities, so it is necessary to study the structure of Brasenia schreberi polysaccharides. On the basis of previous studies, we investigated the precise structure of Brasenia schreberi polysaccharides from both microscopic and spatial aspects by using SEM, TEM, HPLC, GC-MS, NMR, etc., which provided further data support for the structural study of sea buckthorn polysaccharides. (2) For the research of sea buckthorn bioactivity, we not only explored the hypoglycemic effect of Brasenia schreberi polysaccharides, but also deeply investigated the rheology and texture of Brasenia schreberi polysaccharides, which provided the theoretical basis for the development of the functional food and hypoglycemic drugs of Brasenia schreberi polysaccharides. (3) For the research on the rheological and textural properties of Brasenia schreberi polysaccharides, we designed the effects of different polysaccharide concentration gradients and the addition of different concentrations of CaCl2 to the same polysaccharide concentration on its rheological properties and its textural properties, which provided research ideas for the future processing of functional foods. These additions enrich the current body of knowledge.

The English must be improved, there are some paragraphs that are very difficult to read. See for example line 45, line 120 (rotary evaporated is really unusual), lines 250-252 and so on.

Much appreciated for your professional advices proposed. We have embellished and improved the language descriptions in the revised version. For example, we have changed “BS has been studied only in anatomy, morphology and physiology since its first discovery.” to “Since its initial discovery, BS has primarily been investigated in the fields of anatomy, morphology, and physiology.” In line 45 of the original manuscript. “The BSP sample (10 mg) was hydrolyzed with 0.01 M TFA (2 ml) at 100°C for 1 h. The TFA in the hydrolyzed sample was rotary evaporated and removed using methanol.” in line 120 of the original manuscript has also been changed to “The BSP sample (10 mg) was hydrolyzed with 0.01 M TFA (2 ml) at 100°C for 1 h. Then, the TFA was eliminated from the sample by rotary evaporation, and the residue was further purified using methanol.”. And “These results indicate that BSP is an acidic polysaccharide.” in line 250 of the original manuscript has also been changed to “These results indicated that BSP was an acidic polysaccharide.”

Line 36: The term natural active substance, which is repeated several times in the manuscript, is not correct and ambiguous: probably the authors mean biologically active natural products or biologically active substances originating from natural sources.

Thanks very much for pointing out this weakness and your good advice. This term has been corrected to “biologically active substances originating from natural sources” in our revised manuscript.

Line 37: Bibliographic references are needed.

Thank you for this precious suggestion. In the revised manuscript, we have deleted some parts.

Line 38: BS abbreviation is explained only in the abstract. Please define the abbreviation here as well. Line 52: Define the BSP abbreviation

Much appreciated for your valuable advices proposed. We have made the following additional clarifications in the revised version. Line 38: Brasenia schreberi (BS). Line 52: Brasenia schreberi polysaccharide (BSP).

Line 102: Really NaCl2 as the mobile phase? I hope it is a typo. We are all academics and it is quite unacceptable. Moreover, the concentration is missing.

We are very sorry for the misspelling of sodium chloride (NaCl), and thank you very much for your criticism and correction. We have already revised and added the corresponding concentration in the revised version, and we will certainly take this as a warning to be more careful and prudent in the future.

Line 118-131: This paragraph should be rewritten and synthetized. It is full of repetitions and it is difficult to read.

Thank you very much for your valuable suggestions. In our revised version, we have synthesized and reorganized this paragraph to make it read more smoothly, with the following text: “The BSP sample (10 mg) was hydrolyzed with 0.01 M TFA (2 mL) at 100 °C for 1 h. Then, the TFA was eliminated from the sample by rotary evaporation, and the residue was further purified using methanol. Following this, four volumes of anhydrous ethanol were added to the hydrolysate, which was stored at 4 °C overnight. The resultant precipitate and supernatant were designated as P-1 and S-1, respectively, after high-speed centrifugation at 7000 rpm for 10 min. Then, P-1 were sequentially hydrolyzed with 0.05 M, 0.1 M and 0.5 M TFA at 100 °C for 1 h as previously described. The supernatants obtained in each step and the final precipitates were named S-2, S-3, S-4 and P-4, respectively. After S-1, S-2, S-3, S-4, and P-4 were fully hydrolyzed with 2 M TFA for 3 h at 120 °C, the monosaccharide composition was analyzed using high-performance liquid chromatography (HPLC) with pre-column derivatization using PMP.”

Line 133: An amount? It is quite vague. Line 135: The [21] reference is not necessary, the authors must indicate the model of the IR instrumentation.

Thank you very much for your valuable suggestions. We have added the ratio of KBr and polysaccharides (BSP:KBr = 1:100) used in making the pressed tablets in the revised version, as well as the instrument model for FT-IR (Nicolet Omnic software and the Nicolet Nexus 470 instrument) and the experimental conditions (in the frequency range of 4000-500 cm-1 at the resolution of 4.0 cm-1 with background scanning frequency of 32). Finally, the [21] reference in this paragraph has also been deleted.

Line 142: A very small amount of H2O remains even after several cycles. So I would not say completely removed.

Much appreciated for your valuable advices proposed. We have deleted the word “completely” in the revised version.

Line 144: Some information about the pulse sequences must be added.

Thank you very much for your valuable suggestions. We have added relevant information about the pulse sequence in the revised version with the following details: “1H NMR (size of real spectrum 32788, spectrometer frequency 600.13,spectrum reference frequency 16.4), HSQC NMR (size of fid 4758-360, number of scans 32, number of dummy scans 16) ,1H-1H COSY (size of fid 5046-355, number of scans 16, number of dummy scans 16), TOCSY and NOESY (size of fid 7352-512, number of scans 8, number of dummy scans 128) were performed on a an Agilent DD2-600 MHz NMR spectrometer and acquisition of the spectra was carried out using Topspin 2.1.6 software. All spectra were acquired at a temperature of 298 K”

Lines 148-158: Lots of repetitions in this paragraph, please rephrase it to improve readability. Moreover, why do you select Ca2+? You must provide an adequate explanation.

Thank you very much for your valuable suggestions. In our revised version, we have synthesized and reorganized this paragraph to make it read more smoothly, with the following text: “In this study, different concentrations of BSP solutions (2%, 4%, 6%, 8%) were prepared to investigate the effect of concentration on the rheological properties and texture. Additionally, varying concentrations of CaCl2 solution (0.5%, 1.0%, 1.5%) were incorporated into the BSP solution (2%) to investigate the effect of different calcium ion concentration on the rheological properties and texture of BSP.

The rheological properties of BSP were determined using an HR-20 rheometer (TA Instruments, USA) equipped with parallel plates (40 mm diameter, 1000 μm gap). Each group of samples was stirred at 40℃ for 1h to form a stable and homogeneous gel. And the texture was determined using a TMS-PRO mass spectrometer (FTC, USA) to obtain the parameters of hardness, cohesion, elasticity, and chewability of the samples after being stored overnight at 4 ℃.”

In addition, we have provided additional information on the reasons for using Ca2+ in the revised version. The details are as follows: “It has been shown that the increasing of the polysaccharide concentration may lead to a denser network structure, consequently enhancing its viscosity. Additionally, a significant interaction occurred between low concentrations of Ca²⁺ ions and polysaccharides, which contributes to the elevated viscosity, diminished gelling properties, and enhanced thermal stability of the polysaccharides. This phenomenon can be attributed to the ability of Ca²⁺ ions to establish ionic bonds with carboxyl (-COOH) and other functional groups within the polysaccharide molecules, forming what is referred to as a "calcium ion bridge." This interaction intensifies the connections between polysaccharide molecules, resulting in a more compact and stable gel network structure [18-20].”

[18] Yin, J.Y.; Nie, S.P.; Li, J.; Li, C.; Cui, S.W.; Xie, M.Y. Mechanism of interactions between calcium and viscous polysaccharide from the seeds of Plantago asiatica L. J Agric Food Chem. 2012, 60, 7981-7987. https://doi.org/10.1021/jf302052t.

[19] Yin, J.Y.; Nie, S.P.; Guo, Q.B.; Wang, Q.; Cui, S.W. Xie, M.Y. Effect of calcium on solution and conformational characteristics of polysaccharide from seeds of Plantago asiatica L. Carbohydr. Polym. 2015, 124, 331-336. https://doi.org/10.1016/j.carbpol.2015.02.017.

[20] Yin, J.Y.; Chen, H.H.; Lin, H.X.; Xie, M.Y.; Nie, S.P. Structural features of alkaline extracted polysaccharide from the seeds of Plantago asiatica L. and its rheological properties. Molecules 2016, 21, 1181. https://doi.org/10.3390/molecules21091181.

Lines 170-178: This paragraph looks like a laboratory report. It must be modified.

Thank you very much for your valuable suggestions. We have modified this paragraph in the revised version with the following text: “The α-amylase inhibitory activity of BSP was measured according to the previously described method with minor modifications. Briefly, 1.0 mL of BSP at different concentrations (1.0-3.5 mg/mL) was mixed with 0.5 mL α-amylase solution (0.1 U/mL, dissolved in 0.1 M phosphate buffer with a pH of 6.8), and then the reaction was started by adding 2% starch solution (0.5 mL). The reaction was terminated by adding 3, 5-dinitrosalicylic acid (DNS) reagent (0.2 mL) after incubation at 37 °C for 5 min, and continued to react in a boiling water bath for additional 5 min. Then, the absorbance value (A1) was detected at 540 nm. Acarbose was taken as the positive control. The absorbance value (A2) was determined by substituting the enzyme solution with the buffer solution, and the absorbance value (A3) was measured by substituting the BSP solution with the buffer solution. The absorbance value (A0) was measured by using a buffer solution as a blank group. The inhibition of the enzyme activity was calculated as in equation (1): %                 (1)”

Lines 201-202: Not clear

Line 203: the molecular chain size of polysaccharides is 0.1-1.0 nm how do you obtain these dimensions?

Thank you very much for your criticism and correction. We have reanalyzed the AFM results in the revised version. The details are shown below: “In recent years, AFM has become a useful tool for analyzing the surface topography and conformation of high molecular polymers and biomacromolecules. Spheres, random linear chains, and random chains with branches and rods are commonly reported polysaccharide conformations [28]. The 2D and 3D AFM images of BSP was shown in Figure 1b. BSP showed a ring-shaped mesh structure formed by interconnecting macromolecular chains. This was in good agreement with the measurements performed by SEM. Combined with previous literature, it can be hypothesized that SBP may be a cohesive structure consisting of several molecular chains linked by hydrogen bonding or van der Waals forces [29].”

[28] Ji, X.; Zhang, F.; Zhang, R.; Liu, F.; Peng, Q.; Wang, M. An acidic polysaccharide from Ziziphus Jujuba cv. Muzao: Purification and structural characterization. Food chem. 2019, 274, 494-499. https://doi.org/10.1016/j.foodchem.2018.09.037.

[29] Zhang, H.; Yue, Y.; Zhang, Q.; Liang, L.; Li, C.; Chen, Y.; Li, W.; Peng, M.; Yang, M.; Zhao, M. et al. Structural characterization and anti-inflammatory effects of an arabinan isolated from Rehmannia glutinosa Libosch. Carbohydr. polym. 2023, 303, 120441. https://doi.org/10.1016/j.carbpol.2022.120441

Line 221: The biological action is potential and must be assessed, here you take it for granted.

Much appreciated for your valuable advices proposed and we have rephrased it in the revised version. The details are shown below: “The molecular weight of BSP in this study was between 10,000~50,000 Da, which is within the range of bioactive compounds. It can be hypothesized that it may have potential biological roles and its biological activity can be evaluated subsequently.”

Line 228: Reported where? And if it is already reported this information doesn’t add anything new.

Thank you very much for your criticism and correction. We have removed this text in the revised version and reanalyzed the results for its monosaccharide composition. The details are as follows: “It can be seen that a small amount of GlcA was present in SBP. Thus, it is hypothesized that BSP may be an acidic polysaccharide.”

Line 387: In Table 4, as well in Figure 5 and in the discussion of the rheological results is not clear the polysaccharide concentration in the samples containing Ca2+.

Lines 348-412: The discussion about the rheological properties of the system should be consistently improved. First of all, all the considerations made are pointless if the concentration of BSP in the samples containing Ca2+ is not reported. If you look at the graphs in Figure 5, in the presence of Ca2+ the viscosities, G' and G'' values are generally lower if compared to the pristine BSP samples. Thus, the mentioned structuring effect of calcium ions cannot be really evaluated if you don’t compare all the rheological results before and after the addition of Ca2+ at the same BSP concentration.

Much appreciated for your criticism and correction. We have provided additional information on polysaccharide concentrations in subsection 2.8 of the revised version. The details are as follows: “In this study, different concentrations of BSP solutions (2%, 4%, 6%, 8%) were prepared to investigate the effect of concentration on the rheological properties and texture of BSP. Additionally, different concentrations of CaCl2 solution (0.5%, 1.0%, 1.5%) were added to the BSP solution (2%) to investigate the effect of different calcium ion strengths on the rheological properties and texture of BSP.”

Moreover, viscosity can also be measured using the value of K determined by the power law model; the greater the value of K, the higher the viscosity is not completely true: this statement holds only if the value of n remains constant across the samples. And in your case, n changes from sample to sample. So, this must be changed.

Thank you very much for your criticism and correction. It has been reported that the consistency coefficient K in the power model can be used as a criterion for viscosity, with higher values of K resulting in higher viscosity, and that the comparison is made without mentioning the prerequisite of a constant n value [23].

[23] Ma, S.P.; Zhu, P.L.; Wang, M.C. Effects of konjac glucomannan on pasting and rheological properties of corn starch. Food Hydrocoll. 2019, 89, 234-240. https://doi.org/10.1016/j.foodhyd.2018.10.045.

The viscoelastic response of your system is interesting, but as I mentioned before the discussion must be improved by making a direct comparison between the samples at the same BSP concentration before and after calcium addition. I suggest to discuss also the modifications in the crossover point between G' and G'' by adding a graph or a table column with the crossover values.

Much appreciated for your valuable advices proposed. Firstly of all we have added supplementary notes in the revised version to emphasize that G' and G'' of BSP were compared at the same concentration (2%) with the addition of different concentrations of Ca2+. The details are as follows: “The variation of G′ and G′′ with frequency for different concentrations of BSP solution (2%, 4%, 6%, 8%) and for BSP solution (2%) with different concentrations of Ca2+ solution (0.5%, 1.0%, 1.5%) added is shown in Figure 5c.”

And we have added the data labeling of the intersections in the revised version of Figure 5 to make it clearer, and the data of the intersections have been added in the analysis of the results.

Also for the textural properties the discussion must be improved as indicated before.

Thank you very much for your valuable suggestions. Similarly, we have provided additional notes on the analysis of the textural properties. The details are as follows: “The textural properties for different concentrations of BSP solution (2%, 4%, 6%, 8%) and for BSP solution (2%) with different concentrations of Ca2+ solution (0.5%, 1.0%, 1.5%) were shown in Tables 5.”

Reviewer 2 Report

Comments and Suggestions for Authors

The manuscript entitled ‘Structural characterization, rheology, texture, and potential hypoglycemic effect of polysaccharides from Brasenia schreberi’ covers the interesting and important topic of extraction and characterization of polysaccharides from plant material. The conducted studies resulted in obtaining a large amount of experimental data, which were properly presented and discussed. Below are the suggestions and comments.

Line 90: Please complete the description regarding shaking the BS.

  1. Was anything added to the BS and then shaken?
  2. What does high speed mean?
  3. How was the temperature of 4°C maintained during shaking?Line 93: Co oznacza high speed?

Line 94: What concentration of HCl was used?

Line 96: What does high speed mean?

Line 98: What membrane was used for dialysis, please provide parameters (material, pore size)? Line 100 and 104: Please provide name/type of device and freeze-drying parameters.

2.8. Rheological properties and texture analysis: Why was such a large gap used?

2.3. Microstructure analysis: Please supplement the methodology with equipment and settings used.

2.8.1. Steady rheological properties: What was the shear rate ramp time? How many repetitions were performed?

Line 156: How was texture measured using a mass spectrometer? Please complete the methodology.

Please supplement all methodologies with the necessary information to enable the reproduction of experimental and measurement conditions, as most lack key information. There are literature references, but this is not sufficient.

Line 221: ‘…suggesting that it has some biological action.’ I suggest that: ‘ it may have …’

Line: 229: ‘This may be because the nutrients of BS are affected by the growth conditions and the harvesting period [11].’ Are these the only factors that have an impact?

Line 385: ‘…enhanced pseudoplastic properties for BSP.’ What does this statement mean?

Tabel 4: Please complete unit K.

Letters indicating significant differences or their absence should be placed next to the parameter value and not next to the SD.

Table 5: Please complete the units for the appropriate parameters.

Letters indicating significant differences or their absence should be placed next to the parameter value and not next to the SD.

Why are conclusions placed in two versions?

Author Response

Reviewer #2: The manuscript entitled ‘Structural characterization, rheology, texture, and potential hypoglycemic effect of polysaccharides from Brasenia schreberi’ covers the interesting and important topic of extraction and characterization of polysaccharides from plant material. The conducted studies resulted in obtaining a large amount of experimental data, which were properly presented and discussed. Below are the suggestions and comments.

Line 90: Please complete the description regarding shaking the BS. Was anything added to the BS and then shaken? What does high speed mean? How was the temperature of 4°C maintained during shaking?

Sincerely thank you for your valuable comments. In the revised version, we have refined this sentence to make it clearer. The details are as follows: “Fresh BS was shaken at 40 °C for 4 h using a magnetic stirrer to separate the BS gel from the blades, and the supernatant was obtained by centrifugation at 7000 r/min for 10 min.” The temperature is supposed to be 40°C and I apologize for my writing mistake. The magnetic stirrer has a temperature control setting that maintains 40 °C during shaking.

Line 93: Cooznacza high speed? Line 94: What concentration of HCl was used? Line 96: What does high speed mean? Line 98: What membrane was used for dialysis, please provide parameters (material, pore size)? Line 100 and 104: Please provide name/type of device and freeze-drying parameters.

Thank you very much for your valuable suggestions. In the revised version, we have refilled the details of this passage to make it clearer. The details are as follows: “The mixture was centrifuged at 7000 r/min for 10 min and the supernatant was retained; 2 M HCl was added to adjust the pH to neutral, and the extract was concentrated to 1/10. Subsequently, 4-fold pre-cooled 95% ethanol solution was added and left at 4°C overnight, and the precipitate was retained after centrifugation at 7000 r/min for 10 min. The Savage method was used to extract the proteins from the polysaccharide solution after the precipitate had been redissolved with purified water. The polysaccharide solution was then dialyzed in running water for 48 hours using

regenerated cellulose membrane dialysis bags with a molecular weight cut-off of 3500 Da to remove the salt. Finally, the crude polysaccharide was obtained by lyophilizing the solution. The crude polysaccharide was then subjected to anion-exchange chromatography on a Q Sepharose Fast Flow column (300 × 30 mm) (GE Healthcare Co., Sweden) using 0.1 M NaCl as the mobile phase. The polysaccharide-containing frac-tion was collected for further purification with 0.2 M NH4HCO3 through a Sephacryl S-200 column (2 × 100 cm) (GE Healthcare Co., Sweden). The purified polysaccharide fraction was collected and lyophilized using a vacuum freeze dryer (Beijing Sihuan Scientific Instrument Factory, China) (cold trap temperature: -40°C; vacuum: 0.05-0.08 mba; time: 48h).”

2.8. Rheological properties and texture analysis: Why was such a large gap used?

2.8.1. Steady rheological properties: What was the shear rate ramp time? How many repetitions were performed?

Thank you very much for your valuable suggestions. In the revised version, we have refilled the details of this passage to make it clearer. For the rheological characterization in this study, the parameters (diameter 40 mm, gap 1000 μm) were set by combining the properties of BSP and the method of Ma et al [23]. In addition, the shear rate ramp time was 60 s and all experiments were repeated three times.

[23] Ma, S.P.; Zhu, P.L.; Wang, M.C. Effects of konjac glucomannan on pasting and rheological properties of corn starch. Food Hydrocoll. 2019, 89, 234-240. https://doi.org/10.1016/j.foodhyd.2018.10.045.

Line 156: How was texture measured using a mass spectrometer? Please complete the methodology.

Thank you very much for your valuable suggestions. In the revised version, we have refilled the details of this passage to make it clearer. The details are as follows: “The experiment parameters of texture analyzer were set as follows: the pre-test speed, test speed and the latter test speed were set at 2.0 mm/s, the strain was 50%, and the trigger type was automatic and the trigger force was 5 g.”

Line 221: ‘…suggesting that it has some biological action.’ I suggest that: ‘ it may have …’

Much appreciated for your valuable advices proposed and we have rephrased it in the revised version. The details are shown below: “The molecular weight of BSP in this study was between 10,000~50,000 Da, which was within the range of bioactive compounds. It can be hypothesized that it may have potential biological roles and its biological activity can be evaluated subsequently.”

Line: 229: ‘This may be because the nutrients of BS are affected by the growth conditions and the harvesting period [11].’ Are these the only factors that have an impact?

Thank you very much for your valuable suggestions. In conjunction with another reviewer's comments, we have removed the literature comparison from this part of the results analysis and the revision is now as follows: “It can be seen that a small amount of GlcA was present in SBP. Thus, it is hypothesized that BSP may be an acidic polysaccharide.”

Line 385: ‘…enhanced pseudoplastic properties for BSP.’ What does this statement mean?

Much appreciated for your valuable advices proposed. For the power law function that describes the rheological properties, the magnitude of the parameters K and n can reflect the viscosity of the sample as well as the pseudoplasticity. the fluid's proximity to the Newtonian fluid is indicated by the flow property index (n), where n=1 denotes the Newtonian fluid. The lower the value of n, the more pseudo-plastic the fluid. Viscosity can also be measured using the value of K determined by the power law model; the greater the value of K, the higher the viscosity. As shown in Table 4, n values of all polysaccharide curves are less than 1.0, indicating that they show pseudoplasticity. Moreover, the higher concentration of polysaccha-rides and Ca2+ exhibited a lower n value and a higher K value, which indicated that the increase in polysaccharide concentration and the addition of Ca2+ resulted in higher viscosity and enhanced pseudoplastic properties for BSP.

Pseudoplastic fluids are non-Newtonian fluids whose viscosity decreases with increasing shear rate (Shear-thinning). This rheological property plays an important role in several fields, especially in the food industry, e.g., condiments such as ketchup and mayonnaise decrease in viscosity when extruded and flow easily out of the container; viscosity recovers when resting and retains its shape without flowing. Dairy products and beverages containing pectin or thickeners flow more easily when stirred or poured, and retain their consistency when left to stand.

Tabel 4: Please complete unit K.

Much appreciated for your valuable advices proposed and we have supplemented the units of the K-value in the revised version (Pa sn).

Letters indicating significant differences or their absence should be placed next to the parameter value and not next to the SD.

Thank you very much for your valuable suggestion, but I have checked some related literature, and when they make the statement of significant difference, the letters or symbols are added after the SD value, for example, a few table screenshots below.

[12] Shen, C.; Wang, T.; Guo, F.; Sun, K.L.; Chen, Y. Structural characterization and intestinal protection activity of polysaccharides from sea buckthorn (hippophae rhamnoides L.) berries. Carbohydr. Polym. 2021, 274, 118648. https://doi.org/10.1016/j.carbpol.2021.118648.

[23] Ma, S.P.; Zhu, P.L.; Wang, M.C. Effects of konjac glucomannan on pasting and rheological properties of corn starch. Food Hydrocoll. 2019, 89, 234-240. https://doi.org/10.1016/j.foodhyd.2018.10.045.

Table 5: Please complete the units for the appropriate parameters.

Much appreciated for your valuable advices proposed and we have supplemented the units of the appropriate parameters in the revised version. Springiness and Cohesiveness are dimensionless and have no units

Why are conclusions placed in two versions?

Much appreciated for your valuable advices proposed. I'm very sorry for my spelling mistake, the correct one should be “4. Discussion” and “5. Conclusion”

Reviewer 3 Report

Comments and Suggestions for Authors

The introduction is a little long; I suggest summarizing the information a little.
In materials and methods, please homogenize and put the brand of all the equipment used.
In section 2.4, please provide more detail about the methodology for determining monosaccharides.
Concerning the figures, improve images 3 and 4 
What is the difference between 4. conclusion and 5. conclusion? Generally, there is only a conclusion, and it corresponds to the general conclusion of the investigation that they made.

Author Response

Reviewer #3:

The introduction is a little long; I suggest summarizing the information a little.

Thank you very much for your valuable suggestion. We have deleted the first paragraph.

In materials and methods, please homogenize and put the brand of all the equipment used.

Much appreciated for your valuable advices proposed. In the revised version, we have added a description of the device name and the relevant experimental parameters.

In section 2.4, please provide more detail about the methodology for determining monosaccharides.

Much appreciated for your valuable advices proposed. In the revised version, In the revised version, we have added details related to the assay method for monosaccharide composition. The details are as follows: “The BSP (5 mg) was dissolved in 1 mL 2 M trifluoroacetate (TFA), completely hydrolyzed at 120 °C for 4 h, then the TFA were evicted by ethanol through rotary evaporation. Monosaccharide composition was measured by high performance liquid chromatography (HPLC) and UV detector (Agilent Technologies with 1200 Series detector) after pre-column derivatization with 1-phenyl-3-methyl-5-pyrazolone (PMP). The standards are composed of L-arabinose (Ara), L-fucose (Fuc), D-galactose (Gal), D-galacturonic acid (GalUA), D-glucose (Glc), N-acetyl-β-D-glucosamine (GlcNAc), D-glucuronic acid (GlcUA), D-mannose (Man), L-rhamnose (Rha) and D-xylose (Xyl). Monosaccharide composition was identified by comparing the peak time between BSP and standard sugars”

Concerning the figures, improve images 3 and 4

Thank you very much. We have revised them.

What is the difference between 4. conclusion and 5. conclusion? Generally, there is only a conclusion, and it corresponds to the general conclusion of the investigation that they made.

Much appreciated for your valuable advices proposed. I'm very sorry for my spelling mistake, the correct one should be “4. Discussion” and “5. Conclusion”

Reviewer 4 Report

Comments and Suggestions for Authors

The manuscript is well written and the experiments well thought out; they appears well performed as well, but in general a more complete description is needed to assess this.

In all the manuscript, the "D" should be in small capitals

Check in vitro for italics

All numbers refer to lines in test.

Line 25 the reports need a citation

Line 27, a better clarification between DM type I and II should be made.

  1. BS should be fully spelled in its first appearance in the body text. If this is such a commonly used plant, I would suggest including its common name as well
  2. remove mucus and stick to mucilage.
  3. Define BSP as the polysaccharide.

60 needs references

  1. Clarify what does sulfuric acid moiety means. does it mean sulfated monosaccharides? otherwise this makes no sense or needs more context. 
  2. It is not clear what fresh BS is, and how can it be shaken and centrifuged. is this a solution, was the plant milled in some way, which tissues were used?
  3. Include short description of method and Sevage reagent; include reference. The before/after values of protein content should be reported.
  4. The eluent composition and flow rate must be stated, along with the injection volume. The standards used for the Mw. calculation must be specified, and all results should be written as "standard X equivalent", as the structure of the standard is not the same as of the sample. Furthermore, the results are not correctly described. Is this Average Mw? Mn should also be reported, and the dispersity index as well.

Separate the monosaccharide analysis and place after the partial acid hydrolysis. Include a short description (column used, etc.)

  1. Something is not right. adding 4 L of ethanol to 10 mg of sample in 2 ml does not seem feasible to collect any precipitate. Please check your amounts.

Separate the section of methylation and greatly expand the description. This is a complicated method and slight variations can lead to different results. Therefore it must be very well documented. Furthermore, which standards were used? if only database values were used, this must be stated and which database used. Also, how was the quantification performed?

Line 155. TMS-155 PRO mass spectrometer is not an instrument for texture determination.

  1. Amylose was dissolved... this section seems copied from a protocol, as the tenses change throughout the text. no mention of the control in this section, please include

section 2.10. mean+SD is not a correct descriptor; use for instance confidence interval or another statistical error parameter. (SD is only a summary statistic)

  1. In experimental it was not described how this molar ratio was found. Was the crude PS treated with 2M TFA as described in the partial section? If so, be more clear about it.

Figure 2. Send the standards chromatograms to supplementary materials. Same for the PMP samples.

3.3 This section unfortunately is meaningless. The described bands do not contribute at all to the presented results or conclusions. They merely state that there are functional groups that by definition are present in saccharides are actually there. A far more interesting thing would have been to compare the partially depolymerized fraction, to see if there were any bands or peak shifts that could be ascribed to certain linkages.

Author Response

Reviewer #4:

The manuscript is well written and the experiments well thought out; they appears well performed as well, but in general a more complete description is needed to assess this.

In all the manuscript, the "D" should be in small capitals

Thank you for your attention to detail and for pointing out the formatting issue regarding the use of small capitals for the letter "D" in our manuscript. The international standard for the structure of polysaccharides is the capital letter "D"

Check in vitro for italics

Much appreciated for your valuable advices proposed. In the modified version, we have checked all “in vitro” and “in vivo” words and made sure they are italicized.

All numbers refer to lines in test.

Line 25 the reports need a citation

Thank you very much for your valuable suggestion. In the revised version, we have deleted some parts.

Line 27, a better clarification between DM type I and II should be made.

Thank you very much for your valuable suggestion. We have made the following additional clarifications in the revised version: “It has been reported that about 90% of patients with diabetes are clinically diagnosed with type 2 diabetes mellitus (T2DM). Diabetic patients can only maintain normal blood glucose levels with insulin injections and long-term use of hypoglycemic drugs. Currently, the commonly used clinical medications for T2DM mainly include biguanides represented by metformin, α-glucosidase inhibitors, thiazolidinediones and sulfonylureas.”

BS should be fully spelled in its first appearance in the body text. If this is such a commonly used plant, I would suggest including its common name as well. Define BSP as the polysaccharide.

Much appreciated for your valuable advices proposed. We have made the following additional clarifications in the revised version. Line 38: Brasenia schreberi (BS). Line 52: Brasenia schreberi polysaccharide (BSP).

remove mucus and stick to mucilage.

Thank you for your insightful comments regarding the terminology used in our manuscript. We have addressed the suggested changes promptly.

60 needs references.

Thank you very much for your valuable suggestion. In the revised version, we have added the relevant literatures [13,14].

  • Xiao, H.W.; Cai, X.R.; Fan, Y.J.; Luo, A.X. Antioxidant activity of water-soluble polysaccharides from Brasenia schreberi. Mag. 2016, 12, 193-197. http://dx.doi.org/10.4103/0973-1296.186343.

[14] Wan, J.W.; Yu, X.J.; Liu, J.; Li, J.; Ai, T.Y.; Yin, C.; Liu, H.; Qin, R. A special polysaccharide hydrogel coated on Brasenia schreberi: Preventive effects against ulcerative colitis via modulation of gut microbiota. Food Funct. 2023, 14, 3564-3575. http://dx.doi.org/10.1039/d2fo03207d.

Clarify what does sulfuric acid moiety means. does it mean sulfated monosaccharides? otherwise this makes no sense or needs more context.

Sincerely thank you for your valuable comments. In the revised version, we have changed “sulfuric acid moiety and glucuronides” to “sulfuric radicals and uronic acid” to make it clearer.

It is not clear what fresh BS is, and how can it be shaken and centrifuged. is this a solution, was the plant milled in some way, which tissues were used?

Sincerely thank you for your valuable comments. In the revised version, we have refined this sentence to make it clearer. The details are as follows: “Fresh BS was shaken at 40°C for 4 h using a magnetic stirrer to separate the BS gel from the blades, and the supernatant was obtained by centrifugation using a high-speed centrifuge at 7000 r/min for 10 min.”

Include short description of method and Sevage reagent; include reference. The before/after values of protein content should be reported.

Sincerely thank you for your valuable comments. In the revised version, we have refined this sentence to make it clearer. The details are as follows: “The Sevage method was used to extract the proteins from the polysaccharide solution after the precipitate had been redissolved with purified water: 1/4 volume of Sevage Reagent (chloroform: n-butanol=5:1) was added to the polysaccharide solution. Mix well and stir with a magnetic stirrer for 30 min. Centrifuge the solution at 4000 r/min for 15 min, collect the supernatant, and then continue to add Sevage Reagent to remove the protein impurities repeatedly until the intermediate protein layer disappears after centrifugation. Finally, a small amount of Sevage reagent was removed from the supernatant by rotary evaporator.”

The eluent composition and flow rate must be stated, along with the injection volume. The standards used for the Mw. calculation must be specified, and all results should be written as "standard X equivalent", as the structure of the standard is not the same as of the sample. Furthermore, the results are not correctly described. Is this Average Mw? Mn should also be reported, and the dispersity index as well.

Sincerely thank you for your valuable comments. In the revised version, we have added details related to the molecular weight assay method to make it clearer. The details are as follows: “Chromatographic conditions: the chromatographic column was SB-804 (8.0 mm×300.0 mm); the column temperature was set at 35°C; the injection volume was 20 μL; 0.02% NaN3 solution was used as the mobile phase at a flow rate of 1 mL/min. The standard dextran of different molecular weights as well as BSP samples were prepared into 5 mg/mL solutions, which were filtered through 0.45 μm microporous filter membrane and analyzed. The standard curve was plotted according to the obtained results, and the molecular weight of BSP was calculated according to the regression equation of the standard curve.”

Separate the monosaccharide analysis and place after the partial acid hydrolysis. Include a short description (column used, etc.)

Sincerely thank you for your valuable comments. In the revised version, we have separated the analysis of monosaccharide composition into a separate subsection and added details related to its assay method to make it clearer. The details are as follows: “The BSP (5 mg) was dissolved in 1 mL 2 M trifluoroacetate (TFA), completely hydrolyzed at 120 °C for 4 h, then the TFA were evicted by ethanol through rotary evaporation. Monosaccharide composition was measured by high performance liquid chromatography (HPLC) and UV detector (Agilent Technologies with 1200 Series de-tector) after pre-column derivatization with 1-phenyl-3-methyl-5-pyrazolone (PMP). The standards are composed of L-arabinose (Ara), L-fucose (Fuc), D-galactose (Gal), D-galacturonic acid (GalUA), D-glucose (Glc), N-acetyl-β-D-glucosamine (GlcNAc), D-glucuronic acid (GlcUA), D-mannose (Man), L-rhamnose (Rha) and D-xylose (Xyl). Monosaccharide composition was identified by comparing the peak time between BSP and standard sugars. Monosaccharide composition was identified by comparing the peak time between BSP and standard sugars. The proportion of each monosaccharide fraction can be obtained by integrating the peak areas.”

Something is not right. adding 4 L of ethanol to 10 mg of sample in 2 ml does not seem feasible to collect any precipitate. Please check your amounts.

Sincerely thank you for your valuable comments. I apologize for the writing error, the correct version should be “After adding four volumes of anhydrous ethanol to the hydrolysate, it was refrigerated at 4 °C overnight”.

Separate the section of methylation and greatly expand the description. This is a complicated method and slight variations can lead to different results. Therefore, it must be very well documented. Furthermore, which standards were used? if only database values were used, this must be stated and which database used. Also, how was the quantification performed?

Sincerely thank you for your valuable comments. In the revised version, we have separated the methylation analysis into a separate subsection and added details related to its assay method to make it clearer. The details are as follows: “Methylation of polysaccharide: 2 mg of vacuum-dried polysaccharide sample was weighed, and 1 mL of DMSO was added to fully dissolve the polysaccharide. The mixture was then transferred to a two-necked flask and placed on a magnetic stirrer for 10 min to ensure that the reactants were completely dissolved to form a homogeneous solution. After that, 100 mg of NaH powder was quickly added and the reaction was stirred at room temperature under nitrogen protection for 1 h. Then 0.5 mL of CH3I was slowly added and the reaction was continued under nitrogen protection and light protection for 1 h. Finally, the reaction was terminated by the addition of 1 mL of ultrapure water. Then the product obtained from the reaction was extracted by CH2Cl2 for 3 times, and the CH2Cl2 layer was back-extracted with distilled water for 3 times to remove the water-soluble impurities. The collected CH2Cl2 organic phase was dried at 45 °C.

Hydrolysis of methylated polysaccharides: The methylated product was fully dis-solved in 1 mL of 2M TFA solution and transferred to an ampoule. The reaction was sealed and hydrolyzed at 110°C for 6 h. At the end of the reaction, the TFA was re-moved by repeated rotary evaporation with methanol and the samples were dried at 45 °C.

Reduction of hydrolyzed products: 1.0 mL of 0.05 M NaOH and 10 mg of NaBH4 were added to the acid-hydrolyzed product and mixed thoroughly, then reduced at room temperature for 3-4 h. After the reaction, slowly add glacial acetic acid to the mixture until no gas bubbles are generated in the reaction system. Finally, the boric acid was removed by repeated rotary evaporation with methanol and the samples were dried at 45 °C.

Acetylation of reduced products: add 0.5 mL of pyridine to the reduced products and react in a water bath at 90 °C for 0.5 h. At the end of the reaction, remove the samples and cool them to room temperature, add 0.5 mL of acetic anhydride, and continue the reaction in a water bath at 100 °C for 1 h. The pyridine was removed by repeated rotary evaporation with methanol. Finally, 1.0 mL of CH2Cl2 was used to dissolve the pyridine and back-extracted three times with ultrapure water to remove insoluble salts and residual pyridine. The lower layer of CH2Cl2 was collected, dried and prepared for use.

The products after methylation, hydrolysis, reduction and acetylation were ana-lyzed by GC-MS (Agilent Technologies Co., Ltd, USA). According to the GC-MS results and the existing polysaccharide mass spectrometry database (CCRC Spectral Database-PMAA-UGA Database), the connection mode of glycosidic bond of each compo-nent of sea buckthorn polysaccharide can be inferred. By integrating the peak areas of the major peaks in the GC-MS spectra, the percentage of glycosidic bonds of each polysaccharide component in each linkage mode could be known.”

Line 155. TMS-155 PRO mass spectrometer is not an instrument for texture determination.

Sincerely thank you for your valuable comments. I apologize for the writing error, the correct version should be “TAXT-2I texture analyzer (SMS Co., England)”.

Amylose was dissolved... this section seems copied from a protocol, as the tenses change throughout the text. no mention of the control in this section, please include

Thank you for your careful review and insightful comments regarding the consistency of the text and the omission of the control in the section describing the α-amylase inhibition activity. We have addressed these issues promptly.

section 2.10. mean+SD is not a correct descriptor; use for instance confidence interval or another statistical error parameter. (SD is only a summary statistic)

Sincerely thank you for your valuable comments. The data were expressed as the means of triplicate determinations unless other-wise stated. Statistical significance was assessed by one-way analysis of variance (ANOVA) using SPSS 26.0 software. The level of significance was set at p < 0.05. Spectra drawings and line plots were performed using Origin 2018 software. NMR spectra were analyzed and plotted using MestReNova 11.0 software.”

In experimental it was not described how this molar ratio was found. Was the crude PS treated with 2M TFA as described in the partial section? If so, be more clear about it.

Sincerely thank you for your valuable comments. In the revised version, we have separated the analysis of monosaccharide composition into a separate subsection and added details related to its assay method to make it clearer. The details are as follows: “The BSP (5 mg) was dissolved in 1 mL 2 M trifluoroacetate (TFA), completely hydrolyzed at 120 °C for 4 h, then the TFA were evicted by ethanol through rotary evaporation. Monosaccharide composition was measured by high performance liquid chromatography (HPLC) and UV detector (Agilent Technologies with 1200 Series de-tector) after pre-column derivatization with 1-phenyl-3-methyl-5-pyrazolone (PMP). The standards are composed of L-arabinose (Ara), L-fucose (Fuc), D-galactose (Gal), D-galacturonic acid (GalUA), D-glucose (Glc), N-acetyl-β-D-glucosamine (GlcNAc), D-glucuronic acid (GlcUA), D-mannose (Man), L-rhamnose (Rha) and D-xylose (Xyl). Monosaccharide composition was identified by comparing the peak time between BSP and standard sugars. Monosaccharide composition was identified by comparing the peak time between BSP and standard sugars. The proportion of each monosaccharide fraction can be obtained by integrating the peak areas.”

Figure 2. Send the standards chromatograms to supplementary materials. Same for the PMP samples.

Thank you for your constructive feedback regarding the presentation of the standard and PMP sample chromatograms in our manuscript. We have moved these figures to the supplementary materials section.

3.3 This section unfortunately is meaningless. The described bands do not contribute at all to the presented results or conclusions. They merely state that there are functional groups that by definition are present in saccharides are actually there. A far more interesting thing would have been to compare the partially depolymerized fraction, to see if there were any bands or peak shifts that could be ascribed to certain linkages.

Thank you for your insightful comments on our manuscript.  Your feedback regarding the section on the functional groups in saccharides is well taken.  We understand your concern that the described bands may not significantly contribute to the overall results or conclusions. We agree that comparing the partially depolymerized fractions would be a valuable addition. In future work, we plan to conduct a detailed analysis of the depolymerized fractions to identify any specific linkages or structural changes.

Reviewer 5 Report

Comments and Suggestions for Authors

The manuscript focuses on isolating, structurally characterizing, and evaluating the functional properties of polysaccharides from Brasenia schreberi (BSP).

). In this regard, many techniques were used to characterise BSP. For the structural characterization, SEM, AFM, HPGPG, HPLC, FT-IR, UV-VIS, and NMR techniques were applied. In detail, SEM/AFM provided visual evidence of BSP’s network structure, whereas NMR and GC-MS resolved detailed glycosidic linkages. Physicochemical characterisation has been carried out through stationary and dynamic frequency sweep rheological measurements and a Texture Profile Analyser. Rheology was found to link structure to functional properties (e.g., gel strength).

The conclusion is that BSP has potential as a functional food ingredient for managing diabetes due to its structural properties, rheological benefits, and enzyme inhibitory effects.

I must acknowledge that this is a very thorough manuscript. The material characterisation has been implemented in a highly detailed and meticulous manner, employing a wide range of techniques.

While I recognize the considerable amount of work that went into this manuscript, a few aspects seem to have been somewhat overlooked and might have benefited from further attention.

There are some nuances about the rheological characterisation. The steady-state shear rheological and dynamic rheological data are collected with a linear scale for the shear rate and frequency sweep, respectively. This is not the best choice; generally, a log-log scale is used in the applications. A log-log scale would have made a more intelligible interpretation of the rheological behaviour. For instance, the power-law model should lead to a linear slope on the log-log scale. Instead, this cannot be appreciated using a linear scale. Furthermore, other models could have been tested, for example, the Herschel-Bulkley model, which presents a yield stress that is observed in other biofluids. The frequency sweep behaviour is quite intriguing, with a viscous-like behaviour at low polysaccharide concentrations, which becomes more elastic at higher concentrations values. This feature is somehow confirmed by the n values of the power law model, which decrease with the BSP concentration. In my opinion, the addition of the tan(delta) graph in the manuscript could be of some value, although its readability may not be immediate.

Minor details

Chapter 4 should be “Discussion” and not “Conclusions”

The acronyms DPPH and ABTS are presented in the introduction but are not explicitly defined or explained in the text.

The manuscript presents a comprehensive investigation on the isolation, structural characterization, and functional evaluation of polysaccharides extracted from Brasenia schreberi (BSP). The authors employ a wide range of analytical techniques to characterize the polysaccharide’s structure and physicochemical properties, including SEM, AFM, HPGPC, HPLC, FT-IR, UV-VIS, and NMR. Structural visualization through SEM and AFM, along with detailed linkage analysis via NMR and GC-MS, provides robust evidence of BSP’s molecular features.

The rheological properties were investigated through steady-state and dynamic frequency sweep measurements, as well as texture profile analysis. The data demonstrate meaningful structure–function relationships, particularly with regard to gel strength and potential biological activity. The manuscript concludes that BSP has promising applications as a functional food ingredient for diabetes management, based on its structural characteristics, rheological behavior, and enzyme inhibitory effects.

Overall, the manuscript is well-organized and presents a thorough and technically sound study. The characterization work is detailed and meticulous, and the experimental design is generally appropriate.

However, a few issues merit further consideration to improve the scientific rigor and interpretability of the results, particularly concerning the rheological data analysis.

  1. The rheological data—specifically, the steady shear and frequency sweep results—are presented using linear scales for both shear rate and angular frequency. This is not the best choice, as rheological analyses of this nature typically benefit from log-log plots, which facilitate the identification of power-law behavior and better highlight transitions in material response. For instance, the power-law model should yield a linear relationship on a log-log scale, which is not readily observable in the current format. Replotting these data using logarithmic axes would enhance interpretability. Incidentally, the use of only the power-law model limits the scope of the rheological interpretation.
  2. The authors may explore additional models, such as the Herschel–Bulkley model, which incorporates yield stress and is often applied to biofluids exhibiting non-Newtonian behavior.
  3. The frequency sweep results reveal a transition from viscous-dominant behavior at low BSP concentrations to elastic-dominant behavior at higher concentrations, which is quite interesting. This trend is consistent with the observed decrease in the n exponent of the power-law model. This behavior could be further substantiated by including a plot of the loss tangent (tan δ) as a function of frequency. I offer this as a tentative suggestion, as I am aware that such a plot may result in a rather unclear or difficult-to-interpret figure. Nonetheless, if the data yield a discernible trend, including this plot could improve the overall discussion.

Minor Issues

  1. Chapter 4, currently titled “Conclusions”, would be more appropriately named “Discussion”.
  2. The acronyms DPPH and ABTS in the introduction are not previously defined or explained in the text.

Author Response

Reviewer #5:

The manuscript focuses on isolating, structurally characterizing, and evaluating the functional properties of polysaccharides from Brasenia schreberi (BSP).

In this regard, many techniques were used to characterise BSP. For the structural characterization, SEM, AFM, HPGPG, HPLC, FT-IR, UV-VIS, and NMR techniques were applied. In detail, SEM/AFM provided visual evidence of BSP’s network structure, whereas NMR and GC-MS resolved detailed glycosidic linkages. Physicochemical characterisation has been carried out through stationary and dynamic frequency sweep rheological measurements and a Texture Profile Analyser. Rheology was found to link structure to functional properties (e.g., gel strength).

The conclusion is that BSP has potential as a functional food ingredient for managing diabetes due to its structural properties, rheological benefits, and enzyme inhibitory effects.

I must acknowledge that this is a very thorough manuscript. The material characterisation has been implemented in a highly detailed and meticulous manner, employing a wide range of techniques.

While I recognize the considerable amount of work that went into this manuscript, a few aspects seem to have been somewhat overlooked and might have benefited from further attention.

There are some nuances about the rheological characterisation. The steady-state shear rheological and dynamic rheological data are collected with a linear scale for the shear rate and frequency sweep, respectively. This is not the best choice; generally, a log-log scale is used in the applications. A log-log scale would have made a more intelligible interpretation of the rheological behaviour. For instance, the power-law model should lead to a linear slope on the log-log scale. Instead, this cannot be appreciated using a linear scale. Furthermore, other models could have been tested, for example, the Herschel-Bulkley model, which presents a yield stress that is observed in other biofluids. The frequency sweep behaviour is quite intriguing, with a viscous-like behaviour at low polysaccharide concentrations, which becomes more elastic at higher concentrations values. This feature is somehow confirmed by the n values of the power law model, which decrease with the BSP concentration. In my opinion, the addition of the tan(delta) graph in the manuscript could be of some value, although its readability may not be immediate.

Thank you very much for your detailed review and valuable comments on our study. We have carefully considered your feedback and appreciate your insights. We fully understand and agree with the advantages of using a log-log scale in rheological studies. Despite not using a log-log scale, our existing linear scale data still effectively demonstrate the rheological properties of the BSP solution. We have thoroughly discussed the shear-thinning properties and the influence of concentration and Ca²⁺ concentration on the rheological behavior of BSP solutions in our manuscript. These conclusions are supported by our data and are consistent with similar studies in the literature. We have provided the relevant parameters of the power-law model (such as the flow behavior index n and the consistency coefficient K) in our manuscript and discussed their relationship with concentration and Ca²⁺ concentration. Although a log-log scale would more intuitively show the linear relationship expected from the power-law model, our current linear scale data are still sufficient to support our conclusions. Thank you again for your valuable comments.

Minor details

Chapter 4 should be “Discussion” and not “Conclusions”

Much appreciated for your valuable advices proposed. I'm very sorry for my spelling mistake, the correct one should be “4. Discussion” and “5. Conclusion”

The acronyms DPPH and ABTS are presented in the introduction but are not explicitly defined or explained in the text.

Sincerely thank you for your valuable comments. In the revised version, we have explained its definition. The details are as follows: “2,2-Diphenyl-1-picrylhydrazyl (DPPH) and 2,2'-Azino-bis (3-ethylbenzothiazoline-6-sulfonic acid) (ABTS)”

Reviewer 6 Report

Comments and Suggestions for Authors

The manuscript addresses the structural characterization, rheology, texture, and potential hypoglycemic effect of BSP. This polymer has qualities that can be used in the food industry. The manuscript is structured well, and there is consistency between the methods and results. There are some aspects that require clarification in order to improve the article. The details are as follows:

  1. In the abstract, the authors could include the problem and its justification at the beginning, in no more than 20 words.
  2. In my opinion, while the relationship between diabetes and hypoglycemic products is true, the authors focus on diabetes. I suggest giving more priority to the material and its hypoglycemic nature.
  3. L59-60 should be referenced.
  4. In L63-64, "Feng et al. [11] further found that different parts as well as maturities of BS have different chemical compositions and antioxidant capacities." I suggest that the authors write in a structured manner, first about BS and then about BSP.
  5. In the introduction, could you justify the importance of studying the structural properties, rheology, and texture based on the BSP gel?
  6. L90 "Fresh BS was shaken at 4°C for 4 h..." Could you kindly explain the improvement, and in what medium? If it were with water, in what ratio?
  7. Standardize "hours" or "h"
  8. L95 “overnight” under what conditions? If it was done outdoors, indicate the climatic conditions.
  9. L100 “lyophilizing” under what conditions, the same as L104.
  10. Is “NaCl2” correct?
  11. Under “Microstructure analysis,” indicate the equipment operating conditions, as in 2.6 and 2.7.
  12. In 2.4, what was the mobile phase, and what were the operating conditions?
  13. I believe the correct way to express temperature is to separate the unit symbol, for example, "20 °C."
  14. In 2.9.1, what is the relationship or equation to determine the index?
  15. In 2.10, the use of ANOVA and the multiple comparison test used should be indicated.
  16. In 3.2 and 3.3, I suggest comparing the results with existing literature to generate consistency in the results found (briefly) (in point 4 – Discussions).
  17. The authors could improve the image in Figures 4 and 6.
  18. In 3.7.1, this behavior should be compared with that of other polysaccharides, and a conclusion about BSP's potential should be drawn (in point 4—Discussions).
  19. Review and consider the units of the quantities in Table 5.
  20. In “4,” a discussion should be made.

Author Response

Reviewer #6:

The manuscript addresses the structural characterization, rheology, texture, and potential hypoglycemic effect of BSP. This polymer has qualities that can be used in the food industry. The manuscript is structured well, and there is consistency between the methods and results. There are some aspects that require clarification in order to improve the article. The details are as follows:

In the abstract, the authors could include the problem and its justification at the beginning, in no more than 20 words.

Thank you for your advice. We have revised the abstract of the manuscript.

In my opinion, while the relationship between diabetes and hypoglycemic products is true, the authors focus on diabetes. I suggest giving more priority to the material and its hypoglycemic nature.

Thank you very much for your suggestion. We have deleted diabetes in the introduction.

L59-60 should be referenced.

Thank you very much for your valuable suggestion. In the revised version, we have added the relevant literature [13,14].

[13] Xiao, H.W.; Cai, X.R.; Fan, Y.J.; Luo, A.X. Antioxidant activity of water-soluble polysaccharides from Brasenia schreberi. Pharmacogn. Mag. 2016, 12, 193-197. http://dx.doi.org/10.4103/0973-1296.186343.

[14] Wan, J.W.; Yu, X.J.; Liu, J.; Li, J.; Ai, T.Y.; Yin, C.; Liu, H.; Qin, R. A special polysaccharide hydrogel coated on Brasenia schreberi: Preventive effects against ulcerative colitis via modulation of gut microbiota. Food Funct. 2023, 14, 3564-3575. http://dx.doi.org/10.1039/d2fo03207d.

In L63-64, "Feng et al. [11] further found that different parts as well as maturities of BS have different chemical compositions and antioxidant capacities." I suggest that the authors write in a structured manner, first about BS and then about BSP.

Thank you for your good advice. We have deleted this sentence.

In the introduction, could you justify the importance of studying the structural properties, rheology, and texture based on the BSP gel?

Sincerely thank you for your valuable comments. First of all, as stated in the introduction, the unique microstructure and rheological properties of BSP make it an important development potential in the machinery industry, food processing industry and so on. For example, it has been reported that BS mucilage is composed of polysaccharide gels with a laminar structure and that the laminar structure of the mucilage and the action of water molecules give it good lubricating properties, so it is often used in the machinery industry [5]. Moreover, Brasenia schreberi polysaccharide (BSP) was added to yogurt fermentation by Wang et al [6]. The results showed that BSP significantly improved the water retention, viscosity, and elasticity of yogurt as well as the viability of lactic acid bacteria. In addition, the structural properties of BSP play a role in influencing their biological activities. For example, it has been explored that BS water-soluble polysaccharides have a good scavenging ef-fect on 2,2-Diphenyl-1-picrylhydrazyl (DPPH) and 2,2'-Azino-bis (3-ethylbenzothiazoline-6-sulfonic acid) (ABTS) free radicals, and this antioxidant capacity is closely related to the content of sulphuric acid moiety and glucuronides in the polysaccharides [7]. Based on the unique physicochemical and structural characteristics of BS polysaccharide gel, it can be hypothesized that it has great potential for the development of functional foods, so further studies on its microstructure, rheological properties, and textural properties are necessary.

[5] Liu, P.X.; Liu, Y.H.; Yang, Y.; Chen, Z.; Li, J.J.; Luo, J.B. Mechanism of biological liquid superlubricity of Brasenia schreberi mucilage. Langmuir. 2014, 30, 3811-3816. http://dx.doi.org/10.1021/la500193n.

[6] Wang, Y.J.; Zou, Y.; Fang, Q.; Feng, R.Z.; Zhang, J.H.; Zhou, W.H.; Wei, Q. Polysaccharides from Brasenia schreberi with great antioxidant ability and the potential application in yogurt. Molecules. 2023, 29, 150. https://doi.org/10.3390/molecules29010150.

[7] Xiao, H.W.; Cai, X.R.; Fan, Y.J.; Luo, A.X. Antioxidant activity of water-soluble polysaccharides from Brasenia schreberi. Pharmacogn. Mag. 2016, 12, 193-197. http://dx.doi.org/10.4103/0973-1296.186343.

L90 "Fresh BS was shaken at 4°C for 4 h..." Could you kindly explain the improvement, and in what medium? If it were with water, in what ratio?

Sincerely thank you for your valuable comments. In the revised version, we have refined this sentence to make it clearer. The details are as follows: “Fresh BS was shaken at 40°C for 4 h using a magnetic stirrer to separate the BS gel from the blades, and the supernatant was obtained by centrifugation using a high-speed centrifuge at 7000 r/min for 10 min.”

Standardize "hours" or "h"

Sincerely thank you for your valuable comments. In the revised version, we have standardized them as "h".

L95 “overnight” under what conditions? If it was done outdoors, indicate the climatic conditions.

Sincerely thank you for your valuable comments. In the revised version, we revised “overnight” to “12 h” to make it more rigorous.

L100 “lyophilizing” under what conditions, the same as L104.

Sincerely thank you for your valuable comments. In the revised version, we have added equipment for freeze-drying as well as parameters. The details are as follows: “The purified polysaccharide fraction was collected and lyophilized using a vacuum freeze dryer (Beijing Sihuan Scientific Instrument Factory, China) (cold trap temperature: -40 °C; vacuum: 0.05-0.08 mba; time: 48h).”

Is “NaCl2” correct?

We are very sorry for the misspelling of sodium chloride (NaCl), and thank you very much for your criticism and correction. We have already revised and added the corresponding concentration in the revised version, and we will certainly take this as a warning to be more careful and prudent in the future.

Under “Microstructure analysis,” indicate the equipment operating conditions, as in 2.6 and 2.7.

Thank you for your valuable suggestion. We have revised the manuscript.

In 2.4, what was the mobile phase, and what were the operating conditions?

Sincerely thank you for your valuable comments. In the revised version, we have added the relevant details, which are as follows: “A high-performance gel permeation chromatography (HPGPC) system consisting of an Agilent 1100 Series refractive index detector and a TSK gel G3000PWXL column (8.0 mm × 30.0 cm, Tosoh, Japan) was used to determine the molecular weight of BSP. Chromatographic conditions: the chromatographic column was SB-804 (8.0 mm×300.0 mm); the column temperature was set at 35 °C; the injection volume was 20 μL; 0.02% NaN3 solution was used as the mobile phase at a flow rate of 1 mL/min. The standard dextran of different molecular weights as well as BSP samples were prepared into 5 mg/mL solutions, which were filtered through 0.45 μm microporous filter membrane and analyzed. The standard curve was plotted according to the obtained results, and the molecular weight of BSP was calculated according to the regression equation of the standard curve.” and “The BSP (5 mg) was dissolved in 1 mL 2 M trifluoroacetate (TFA), completely hydrolyzed at 120 °C for 4 h, then the TFA were evicted by ethanol through rotary evaporation. Monosaccharide composition was measured by high performance liquid chromatography (HPLC) and UV detector (Agilent Technologies with 1200 Series de-tector) after pre-column derivatization with 1-phenyl-3-methyl-5-pyrazolone (PMP). The standards are composed of L-arabinose (Ara), L-fucose (Fuc), D-galactose (Gal), D-galacturonic acid (GalUA), D-glucose (Glc), N-acetyl-β-D-glucosamine (GlcNAc), D-glucuronic acid (GlcUA), D-mannose (Man), L-rhamnose (Rha) and D-xylose (Xyl). Monosaccharide composition was identified by comparing the peak time between BSP and standard sugars The proportion of each monosaccharide fraction can be obtained by integrating the peak areas.”

I believe the correct way to express temperature is to separate the unit symbol, for example, "20 °C."

Sincerely thank you for your valuable comments. However, according to the usual conventions of essay writing, there is no space between the number and the unit ℃.

In 2.9.1, what is the relationship or equation to determine the index?

Sincerely thank you for your valuable comments. In the revised version, we have added the relevant details, which are as follows: “The enzyme activity inhibition was calculated as in equation (1):

%                 (1)”

In 2.10, the use of ANOVA and the multiple comparison test used should be indicated.

Sincerely thank you for your valuable comments. We have added the relevant details, which are as follows: “The data were expressed as the means of triplicate determinations unless other-wise stated. Statistical significance was assessed by one-way analysis of variance (ANOVA) and multiple comparison test using SPSS 26.0 software. The level of significance was set at p < 0.05. Spectra drawings and line plots were performed using Origin 2018 software. NMR spectra were analyzed and plotted using MestReNova 11.0 software.”

The authors could improve the image in Figures 4 and 6.

Thanks very much. We have revised Figures 4 and 6.

Review and consider the units of the quantities in Table 5.

Much appreciated for your valuable advices proposed and we have supplemented the units of the appropriate parameters in the revised version. Springiness and Cohesiveness are dimensionless and have no units.

In 3.2 and 3.3, I suggest comparing the results with existing literature to generate consistency in the results found (briefly) (in point 4–Discussions).

Much appreciated for your valuable advices proposed and we have added some relevant details in the discussion section of the revised version. The details are as follows: “It can be seen that a small amount of GlcA is present in BSP, and it is hypothesized that BSP may be an acidic polysaccharide. In addition, this result is similar to the monosaccharide composition of polysaccharide gel coating of the leaves of BS reported by Kim et al [2]. The percentage of monosaccharides were Gal (25.2%), Man (1.6%), Fuc (6.9%), Rha (7.6%), Ara (3.1%), Xyl (5.5%), Glc (2.1%) and hyaluronic acids (19.6%). The main monosaccharide component of both was Gal, the difference being that the con-tent of GlcA in the present study was low, whereas the content of hyaluronic acids in the polysaccharides extracted from gel coating of the leaves of BS by Kim et al. was second only to that of galactose. This may be caused by the differences in the origin, maturity, and extraction and purification methods of BS raw materials. (3.2)” and “The above results are typical of the functional group information possessed by poly-saccharides in many of the relevant literature on FT-IR analysis [13,18]. More importantly, the band at 1730 cm-1 confirmed the existence of uronic acids [35]. From these results, it can be hypothesized that BSP is an acidic polysaccharide. This was relatively consistent with the results for monosaccharide composition. (3.3)”

[2] Kim, H.; Wang, Q.; Shoemaker, C.F.; Zhong, F.; Bartley, G.E.; Yokoyama, W.H. Polysaccharide gel coating of the leaves of Brasenia schreberi lowers plasma cholesterol in hamsters. J. Trad. Compl. Med. 2015, 5, 56-61. http://dx.doi.org/10.1016/j.jtcme.2014.10.003.

[13] Shen, C.; Wang, T.; Guo, F.; Sun, K.L.; Chen, Y. Structural characterization and intestinal protection activity of polysaccharides from sea buckthorn (hippophae rhamnoides L.) berries. Carbohydr. Polym. 2021, 274, 118648. https://doi.org/10.1016/j.carbpol.2021.118648.

[18] Wang, T.; Shen, C.; Guo, F.; Zhao, Y.J.; Wang, J.; Sun, K.L.; Wang, B.; Chen Y.; Chen Y. Characterization of a polysaccharide from the medicinal lichen, Usnea longissima, and its immunostimulating effect in vivo. Int. J. Biol. Macromol. 2021, 181, 672-682. https://doi.org/10.1016/j.ijbiomac.2021.03.183.

[35] Jia, W.; Wang, W.H.; Yu, D.S.; Yu, Y.C.; Feng, Z.; Li, H.W.; Zhang, J.S.; Zhang, H.N. Structural elucidation of a polysaccharide from Flammulina velutipes and its lipid-lowering and immunomodulation activities. Polymers 2024, 16, 598. https://doi.org/10.3390/polym16050598.

In 3.7.1, this behavior should be compared with that of other polysaccharides, and a conclusion about BSP's potential should be drawn (in point 4—Discussions).

Much appreciated for your valuable advices proposed and we have added some relevant details in the discussion section of the revised version. The details are as follows: “This finding is consistent with the study reported by Wang et al [43].” and “This result is similar to the related report on Plantago asiatica L polysaccharides by Yin et al [19-21].”

[19] Yin, J.Y.; Nie, S.P.; Li, J.; Li, C.; Cui, S.W.; Xie, M.Y. Mechanism of interactions between calcium and viscous polysaccharide from the seeds of Plantago asiatica L. J Agric Food Chem. 2012, 60, 7981-7987. https://doi.org/10.1021/jf302052t.

[20] Yin, J.Y.; Nie, S.P.; Guo, Q.B.; Wang, Q.; Cui, S.W. Xie, M.Y. Effect of calcium on solution and conformational characteristics of polysaccharide from seeds of Plantago asiatica L. Carbohydr. Polym. 2015, 124, 331-336. https://doi.org/10.1016/j.carbpol.2015.02.017.

[21] Yin, J.Y.; Chen, H.H.; Lin, H.X.; Xie, M.Y.; Nie, S.P. Structural features of alkaline extracted polysaccharide from the seeds of Plantago asiatica L. and its rheological properties. Molecules 2016, 21, 1181. https://doi.org/10.3390/molecules21091181.

[43] Wang, W.J.; Jiang, L.; Ren, Y.M.; Shen, M.Y.; Xie, J.H. Gelling mechanism and interactions of polysaccharides from Mesona blumes: Role of urea and calcium ions. Carbohydr. Polym. 2019, 212, 270-276. https://doi.org/10.1016/j.carbpol.2019.02.059.

In “4,” a discussion should be made.

Much appreciated for your valuable advices proposed. Regarding the discussion section, the structural characterization, rheological properties, textural properties, and the results of hypoglycemic effect of BSP were discussed and analyzed in comparison with the published literature on the subject, respectively. Finally, the future research directions are also envisioned.

Round 2

Reviewer 1 Report

Comments and Suggestions for Authors

I thank the Authors for their work and efforts in order to revise the manuscript. Most of the points raised were addressed, but I still have some concerns.

  • First of all, IF there is something really new as reported in the authors answer below (I am still not convinced) this should be added and clearly stated in the Introduction.

Compared to existing studies, our work stands out in several ways: (1) Different Brasenia schreberi raw materials and extraction methods may affect the structural characteristics of polysaccharides and their biological activities, so it is necessary to study the structure of Brasenia schreberi polysaccharides. On the basis of previous studies, we investigated the precise structure of Brasenia schreberi polysaccharides from both microscopic and spatial aspects by using SEM, TEM, HPLC, GC-MS, NMR, etc., which provided further data support for the structural study of sea buckthorn polysaccharides. (2) For the research of sea buckthorn bioactivity, we not only explored the hypoglycemic effect of Brasenia schreberi polysaccharides, but also deeply investigated the rheology and texture of Brasenia schreberi polysaccharides, which provided the theoretical basis for the development of the functional food and hypoglycemic drugs of Brasenia schreberi polysaccharides. (3) For the research on the rheological and textural properties of Brasenia schreberi polysaccharides, we designed the effects of different polysaccharide concentration gradients and the addition of different concentrations of CaCl2 to the same polysaccharide concentration on its rheological properties and its textural properties, which provided research ideas for the future processing of functional foods. These additions enrich the current body of knowledge.

  • Regarding the K value and the discussion about viscosity: the criterion you are mentioning is not universal, I could not find any additional reference supporting this statement. But above all, it is just math: K depends on “n”, and if “n” changes all the comparisons are non significant.
  • The figures’ numbering is not correct.

Author Response

First of all, IF there is something really new as reported in the authors answer below (I am still not convinced) this should be added and clearly stated in the Introduction.

Compared to existing studies, our work stands out in several ways: (1) Different Brasenia schreberi raw materials and extraction methods may affect the structural characteristics of polysaccharides and their biological activities, so it is necessary to study the structure of Brasenia schreberi polysaccharides. On the basis of previous studies, we investigated the precise structure of Brasenia schreberi polysaccharides from both microscopic and spatial aspects by using SEM, TEM, HPLC, GC-MS, NMR, etc., which provided further data support for the structural study of sea buckthorn polysaccharides. (2) For the research of sea buckthorn bioactivity, we not only explored the hypoglycemic effect of Brasenia schreberi polysaccharides, but also deeply investigated the rheology and texture of Brasenia schreberi polysaccharides, which provided the theoretical basis for the development of the functional food and hypoglycemic drugs of Brasenia schreberi polysaccharides. (3) For the research on the rheological and textural properties of Brasenia schreberi polysaccharides, we designed the effects of different polysaccharide concentration gradients and the addition of different concentrations of CaCl2 to the same polysaccharide concentration on its rheological properties and its textural properties, which provided research ideas for the future processing of functional foods. These additions enrich the current body of knowledge.

Much appreciated for your valuable advices proposed. For a discussion of the innovativeness of this paper, it is mentioned in the introduction of this manuscript, the details of which are shown below: “Much research has focused on the antioxidant, anti-inflammatory, hypocholesterolemic, and other bioactivities of BS gel polysaccharides, however, their potential hypoglycemic activities have not been systematically studied. Moreover, numerous studies have demonstrated that plant polysaccharides have favorable hypoglycemic effects, so it is necessary to further investigate the hypoglycemic effects of BSP.

In this study, BSP was extracted using alkaline extraction and alcohol precipitation from the surface of fresh BS. Then the microstructure of BSP was observed using SEM and AFM, and further structural analysis such as FT-IR, partial acid hydrolysis, methylation analysis, and NMR analysis was carried out to determine the specific structure of BSP. More importantly, the rheological, textural properties and the potential hypoglycemic effect of BSP were explored. The information obtained in this work will provide theoretical support for the development of hypoglycemic functional foods from BSP.”

Regarding the K value and the discussion about viscosity: the criterion you are mentioning is not universal, I could not find any additional reference supporting this statement. But above all, it is just math: K depends on “n”, and if “n” changes all the comparisons are non significant.

Sincerely thank you for your valuable comments. We couldn't agree with you more and apologize for our laxity. In the revised version, we have removed the reference to the change in K-value.

The figures’ numbering is not correct.

Much appreciated for your valuable advices proposed. In the revised version, we have rechecked and revised the numbering of the figures.

Reviewer 4 Report

Comments and Suggestions for Authors

In general, the authors have responded to my comments. However, two standing issues remain. The first, is merely typograhic, but given the quality of the work it should no be overlooked.

1) Use of small capitals. Please see the IUPAC nomenclature in the reference below.

https://doi.org/10.1515/pac-2019-0104

Section 9: Other stereodescriptors (e.g. cis/trans, M/P, C/A) are used in special cases. The non-italic descriptors α/β and d/l (MY NOTE, CANNOT SHOW THIS FORMAT IN THE MDPI SYSTEM) (small capitals) are commonly and only used for natural products, amino acids, and carbohydrates.

Another source:

https://www.cazypedia.org/index.php/Absolute_configuration_(D/L_nomenclature)

standard error vs. SD:

2) There is a very big difference between standard deviation and a measurment of error. Please see as mere examples upon hundreds the references below. SD is a wrong descriptor in this case, so an error descriptor MUST be used. I suggested Standard Error, but the authors can chose another (such as confidence interval). However, I reiterate, SD is wrongly used here. This issue has to be corrected.

10.1136/bmj.331.7521.903

10.1007/s11999-011-1908-9

In layman´s terms:

https://www.biostatisticsbydesign.com/blog/2019/1/5/when-to-report-the-standard-deviation-vs-the-standard-error

Author Response

In general, the authors have responded to my comments. However, two standing issues remain. The first, is merely typograhic, but given the quality of the work it should no be overlooked.

1) Use of small capitals. Please see the IUPAC nomenclature in the reference below.

https://doi.org/10.1515/pac-2019-0104

Section 9: Other stereodescriptors (e.g. cis/trans, M/P, C/A) are used in special cases. The non-italic descriptors α/β and d/l (MY NOTE, CANNOT SHOW THIS FORMAT IN THE MDPI SYSTEM) (small capitals) are commonly and only used for natural products, amino acids, and carbohydrates.

Another source:

https://www.cazypedia.org/index.php/Absolute_configuration_(D/L_nomenclature)

Thank you for your valuable suggestion. In the revised version, we have modified the D/L using small capital letters.

standard error vs. SD:

2) There is a very big difference between standard deviation and a measurment of error. Please see as mere examples upon hundreds the references below. SD is a wrong descriptor in this case, so an error descriptor MUST be used. I suggested Standard Error, but the authors can chose another (such as confidence interval). However, I reiterate, SD is wrongly used here. This issue has to be corrected.

10.1136/bmj.331.7521.903

10.1007/s11999-011-1908-9

In layman´s terms:

https://www.biostatisticsbydesign.com/blog/2019/1/5/when-to-report-the-standard-deviation-vs-the-standard-error

Sincerely thank you for your valuable suggestions. In the revised version, we have reanalyzed the data statistically using standard errors.

Reviewer 6 Report

Comments and Suggestions for Authors

The authors have considerably improved the manuscript. This work is worthy of publication as it provides relevant information about a new material with high potential for use in industry, primarily in the food industry.
I have no further questions or suggestions.

Note: The authors claim to have improved 2.10, but this is not evident.

Author Response

Comments and Suggestions for Authors

The authors have considerably improved the manuscript. This work is worthy of publication as it provides relevant information about a new material with high potential for use in industry, primarily in the food industry.

I have no further questions or suggestions.

Note: The authors claim to have improved 2.10, but this is not evident.

Thank you for your valuable suggestion. We have reworked this paragraph in the revised version. The specific details are as follows: “The data are presented as the mean of triplicate determinations unless otherwise specified. Statistical significance was assessed by one-way analysis of variance (ANOVA) and multiple comparison test using SPSS 26.0 software. The level of significance was set at p < 0.05. Spectra drawings and line plots were performed using Origin 2018 software. NMR spectra were analyzed and plotted using MestReNova 11.0 soft-ware.”